# Optimal Attention Temperature Enhances In-Context Learning under Distribution Shift

## Abstract

Pretrained Transformers excel at in-context learning (ICL), inferring new tasks from only a handful of examples. Yet, their ICL performance can degrade sharply under distribution shift between pretraining and test data—a regime increasingly common in real-world deployments. While recent empirical work hints that adjusting the attention temperature in the softmax can enhance Transformer performance, the attention temperature's role in ICL under distribution shift remains unexplored. This paper provides the first theoretical and empirical study of attention temperature for ICL under distribution shift. Using a simplified but expressive "linearized softmax" framework, we derive closed-form generalization error expressions and prove that shifts in input covariance or label noise substantially impair ICL, but that an optimal attention temperature exists which minimizes this error. We then validate our predictions through extensive simulations on linear regression tasks and large-scale experiments with GPT-2 and LLaMA2-7B on question-answering benchmarks. Our results establish attention temperature as a principled and powerful mechanism for improving the robustness of ICL in pretrained Transformers—advancing theoretical understanding and providing actionable guidance for selecting attention temperature in practice.

## 1 Introduction

Transformers (Vaswani et al., 2017) have emerged as the foundational architecture of contemporary AI systems, underpinning state-of-the-art models such as ChatGPT, Gemini, and DeepSeek. Central to their remarkable success is *in-context learning* (ICL)—the capability to adapt to novel tasks directly from prompts without any gradient-based weight updates (Brown et al., 2020). This property, often described as emergent, has catalyzed a surge of research aimed at uncovering the underlying mechanisms of ICL (Akyürek et al., 2023; Von Oswald et al., 2023) and at characterizing how factors such as task diversity and model scale shape its performance (Wei et al., 2022; Wu et al., 2024).

Yet, despite its transformative potential, ICL exhibits pronounced sensitivity to distributional shifts between pretraining and downstream tasks. Both empirical and theoretical investigations demonstrate that even mild shifts can substantially degrade performance (Zhang et al., 2024), underscoring unresolved questions about the robustness, generalization, and adaptability of pretrained Transformer models. Addressing these limitations is crucial for realizing the full promise of ICL in reliable, deployable AI systems.

At the core of the Transformer architecture is the self-attention mechanism, defined as

$$\text{Attention}(\boldsymbol{Z}) := \boldsymbol{VZ} \cdot \text{softmax}\left(\frac{(\boldsymbol{KZ})^T(\boldsymbol{QZ})}{\tau}\right), \tag{1}$$

where $\boldsymbol{Z}$ denotes the input representation and $\boldsymbol{Q}$, $\boldsymbol{K}$, and $\boldsymbol{V}$ are the query, key, and value weight matrices, respectively. The parameter $\tau > 0$, referred to as the *attention temperature*, controls the variance of the softmax outputs and hence the selectivity of attention weights. This quantity is distinct from the "sampling temperature" commonly used to adjust the output distribution of generative models such as large language models (LLMs) (Renze & Guven, 2024). Throughout this work, we exclusively focus on the attention temperature as an intrinsic component of the attention mechanism.

While the original Transformer fixes $\tau = \sqrt{d_k}$ (Vaswani et al., 2017), where $d_k$ is the key dimension, subsequent empirical studies have demonstrated that adjusting the attention temperature can enhance performance across diverse NLP and computer vision benchmarks (Lin et al., 2018; Zhang et al., 2022; Peng et al., 2024; Lee et al., 2021; Chen et al., 2023; Zou et al., 2024). Yet, to the best of our knowledge, its role within *in-context learning* (ICL) remains unexplored. Because the attention temperature directly governs how sharply the model concentrates on specific inputs, it is poised to critically influence ICL behavior under distributional shift—a setting of central practical relevance, where mismatches between training and inference distributions are ubiquitous.

**This work —** In this paper, we present a unified theoretical and empirical study of the *attention temperature* in the context of in-context learning (ICL). We focus on how adjusting this parameter can systematically improve the ICL performance of pretrained Transformers under distributional shift. We address this question in the setting of linear regression tasks, which offer a well-controlled yet expressive framework for dissecting the mechanisms of ICL (Garg et al., 2022; Zhang et al., 2024). Departing from prior work restricted to linear attention, we analyze a Transformer with *linearized softmax* attention—an architecture that preserves the essential temperature-dependent behavior of standard attention while remaining mathematically tractable.

Our analysis yields a closed-form characterization of the *optimal temperature*—the value of $\tau$ that minimizes generalization error during inference. We show that this optimal temperature depends explicitly on the nature of the distribution shift and that setting it appropriately can recover or even surpass baseline ICL performance. We validate our theoretical predictions through extensive experiments on both synthetic (linear regression) and real-world (question answering with LLMs) tasks, demonstrating that temperature selection constitutes a simple yet powerful mechanism for improving robustness.

**Contributions —** Our work makes the following key contributions:

1. We provide, to our knowledge, the first theoretical characterization of the optimal attention temperature for pretrained Transformers with *linearized softmax attention* in ICL tasks.
2. We analyze the generalization behavior of such models under a broad spectrum of distributional shifts, employing weaker assumptions than prior studies.
3. We establish a clear theoretical and empirical link between distribution shift and attention temperature, showing that principled temperature selection can substantially enhance ICL performance across diverse tasks.

Taken together, these results offer new insights into the interplay between temperature, distribution shift, and generalization in in-context learning, and highlight a practical avenue for improving the robustness of pretrained Transformers.

## 2 RELATED WORK

**Theory of in-context learning —** Simplified Transformer variants—particularly those using linear attention—have proven useful for gaining analytical insights about ICL (Garg et al., 2022; Zhang et al., 2024; Raventós et al., 2023). Notably, Zhang et al. (2024) showed that linear Transformers approximate Bayes-optimal inference in linear regression tasks, even under distribution shift. We build on this line of research but focus explicitly on the role of the *attention temperature*. In contrast to Zhang et al. (2024), we (i) employ *linearized softmax attention* to isolate the effect of temperature, (ii) study how temperature adjustments can mitigate the impact of distribution shifts, and (iii) derive and empirically evaluate the *optimal temperature* for improving ICL performance. These advances extend prior analyses and yield a deeper theoretical and empirical understanding of how principled temperature selection enhances the robustness of Transformers under distributional shift.[1]

**Linear vs. softmax attention —** Although linear attention has gained traction for its computational efficiency, it typically lags behind softmax-based counterparts in predictive performance, spurring efforts to narrow this gap (Choromanski et al., 2021; Qin et al., 2022). A pivotal advance in this direction is due to Han et al. (2024), who showed that a *linearized variant of softmax attention* can closely approximate the performance of standard softmax attention. Building on this insight, we adopt the *linearized softmax* formulation, which preserves the essential temperature-dependent behavior of standard attention while enabling tractable theoretical analysis. This choice provides a

---

[1]Due to space limitations, additional related work is discussed in Appendix L.

principled framework for investigating how attention-temperature selection shapes ICL performance in pretrained Transformers.

**Attention temperature** — Research on attention temperature remains limited. Veličković et al. (2025) recently proposed an adaptive temperature scheme to sharpen softmax outputs, and several empirical studies in natural language processing and computer vision (Lin et al., 2018; Zhang et al., 2022; Peng et al., 2024; Lee et al., 2021; Chen et al., 2023; Zou et al., 2024) suggest that adjusting the attention temperature can enhance Transformer performance. However, these works do not examine ICL under distributional shift. To our knowledge, no prior study has systematically analyzed how attention temperature influences ICL in such settings—a gap our work directly addresses.

# 3 SETTING

We describe the setup for analyzing ICL in linear regression using pretrained Transformers, covering the data model, linearized attention with reparameterization, evaluation metrics, and the Bayes-optimal benchmark.

**Notation** — We follow standard notation from Goodfellow et al. (2016). The spectral norm of matrix $\boldsymbol{M}$ is denoted by $\|\boldsymbol{M}\|$, and the trace by $\mathrm{Tr}(\boldsymbol{M})$. Matrix entries and slices are denoted as $M_{i,j}$, $\boldsymbol{M}_{:,j}$, and $\boldsymbol{M}_{i,:}$.

## 3.1 PROBLEM SETUP: IN-CONTEXT LEARNING FOR LINEAR REGRESSION

We study the ICL abilities of pretrained Transformers on linear regression tasks. Given a sequence of tokens, i.e., input-label pairs, $\{(\boldsymbol{x}_1, y_1), (\boldsymbol{x}_2, y_2), \ldots, (\boldsymbol{x}_{l-1}, y_{l-1}), (\boldsymbol{x}_l, ?)\}$ where each input vector $\boldsymbol{x}_i \in \mathbb{R}^d$ and corresponding label $y_i \in \mathbb{R}$ are independently sampled from an unknown joint distribution, the model must predict $y_l$ using only the context $\{(\boldsymbol{x}_i, y_i)\}_{i=1}^{l-1}$ and the query $\boldsymbol{x}_l$, where $l-1$ is referred as the "context length". Each $(\boldsymbol{x}_i, y_i)$ pair is sampled i.i.d. from a joint distribution defined by:

$$\boldsymbol{x}_i \sim \mathcal{N}(\boldsymbol{\mu}_x, \boldsymbol{\Sigma}_x), \quad y_i = \boldsymbol{w}^T \boldsymbol{x}_i + \epsilon_i, \quad \epsilon_i \sim \mathcal{N}(0, \sigma^2), \tag{2}$$

where the task vector $\boldsymbol{w} \sim \mathcal{N}(\boldsymbol{\mu}_w, \boldsymbol{\Sigma}_w)$ is fixed within a context but varies across tasks.

**Assumption 3.1** (Well-Behaved Data Distributions). There exist constants $c_1, c_2, c_3 > 0$ such that:

$$\|\boldsymbol{\mu}_x\|, \|\boldsymbol{\mu}_w\| \leq c_1, \quad \lambda_{\min}(\boldsymbol{\Sigma}_x), \lambda_{\min}(\boldsymbol{\Sigma}_w) \geq c_2, \quad \lambda_{\max}(\boldsymbol{\Sigma}_x), \lambda_{\max}(\boldsymbol{\Sigma}_w) \leq c_3.$$

This assumption ensures that the input and task distributions have bounded means and covariances, offering greater flexibility than the more restrictive setup of Zhang et al. (2024).

**Assumption 3.2** (High-Dimensional Regime). The context length $l$ and input dimension $d$ diverge jointly: $l, d \to \infty$.

This assumption reflects realistic settings where both context length and input dimension grow simultaneously, aligning with modern ML trends and enabling analysis of generalization in high-dimensional regimes.

Under this set of assumptions, we define ICL for linear regression tasks as follows:

**Definition 3.3** (In-Context Learning (ICL)). A model succeeds at ICL for linear regression if its generalization error nearly matches that of the Bayes-optimal linear model (defined in Section 3.6).

## 3.2 MODELING ATTENTION WITH TRANSFORMERS

Following the convention established by Zhang et al. (2024), we represent the input sequence by an embedding matrix:

$$\boldsymbol{Z} := \begin{bmatrix} \boldsymbol{x}_1 & \cdots & \boldsymbol{x}_{l-1} & \boldsymbol{x}_l \\ y_1 & \cdots & y_{l-1} & 0 \end{bmatrix} \in \mathbb{R}^{(d+1) \times l}, \tag{3}$$

where the last column corresponds to the query input with no label.

Given this embedding, the softmax self-attention output is

$$\boldsymbol{S} := \boldsymbol{Z} + \boldsymbol{V}\boldsymbol{Z} \cdot \text{softmax}\left(\frac{(\boldsymbol{K}\boldsymbol{Z})^T(\boldsymbol{Q}\boldsymbol{Z})}{\tau}\right), \tag{4}$$

where $\boldsymbol{K}$, $\boldsymbol{Q}$, and $\boldsymbol{V}$ are the key, query, and value matrices, respectively, and $\tau$ is the temperature.

Here, we denote the model's prediction as $S_{d+1,l}$ — the last element in the final row.

## 3.3 LINEARIZED ATTENTION APPROXIMATION

To analytically characterize the effect of temperature on ICL, we employ a linearized approximation of softmax attention (see Appendix B for the derivation and formal definition):

$$\boldsymbol{E} := \boldsymbol{Z} + \frac{1}{l}\boldsymbol{V}\boldsymbol{Z}\left(\frac{(\boldsymbol{K}\boldsymbol{Z})^T(\boldsymbol{Q}\boldsymbol{Z})}{\tau} + \mathbf{1} - \frac{1}{l}\sum_{j=1}^{l}\frac{(\boldsymbol{K}\boldsymbol{Z}_{:,j})^T(\boldsymbol{Q}\boldsymbol{Z})}{\tau}\right), \tag{5}$$

where $\hat{y} := E_{d+1,l}$ represents the predicted label. In contrast to linear attention (Zhang et al., 2024),

$$\boldsymbol{Z} + \frac{1}{l}\boldsymbol{V}\boldsymbol{Z}(\boldsymbol{K}\boldsymbol{Z})^T(\boldsymbol{Q}\boldsymbol{Z}), \tag{6}$$

our formulation in (5) explicitly preserves normalization, which is essential for both interpretability and robustness. This difference is described in the following remark.

*Remark* 3.4 (Linear vs. linearized attention). Linearized attention maintains row-wise normalization, making it inherently more robust to shifts in input means — a critical failure mode of linear attention in ICL. Appendix C provides an illustrative comparison.

Another key distinction between linear case and linearized softmax case is that linear (with temperature scaling) fails to capture the temperature behavior of softmax. However, while this may not be immediately apparent, linearized softmax closely mirrors the behavior of softmax with respect to temperature variation. A detailed explanation together with an illustrative example is provided in Appendix D.

## 3.4 REPARAMETRIZATION OF LINEARIZED ATTENTION

To streamline analysis, we reparametrize the matrices $\boldsymbol{V}$ and $\boldsymbol{M} := \boldsymbol{K}^T\boldsymbol{Q}$ as:

$$\boldsymbol{V} = \begin{bmatrix} * & * \\ \boldsymbol{v}_{21}^T & v_{22} \end{bmatrix}, \quad \boldsymbol{M} = \begin{bmatrix} \boldsymbol{M}_{11} & * \\ \boldsymbol{m}_{21}^T & * \end{bmatrix}, \tag{7}$$

where only $\boldsymbol{v}_{21}$, $v_{22}$, $\boldsymbol{m}_{21}$, and $\boldsymbol{M}_{11}$ influence the prediction $\hat{y}(\boldsymbol{Z}; \boldsymbol{V}, \boldsymbol{M})$. The remaining terms are denoted by $*$ as they are not relevant for predicting $y_l$ in this context. The prediction from the Transformer model (5) can thus be expressed as a function of $\boldsymbol{M}$ and $\boldsymbol{V}$, i.e., $\hat{y}(\boldsymbol{Z}; \boldsymbol{V}, \boldsymbol{M}) := E_{d+1,l}$. This form parallels the approach by Zhang et al. (2024), allowing for tractable theoretical analysis.

By analyzing this reparameterization, we gain a deeper understanding of how the model parameters interact with the data to address the ICL problem effectively. This foundational insight will provide the necessary basis for discussing the pretraining of these parameters in Section 4.1.

## 3.5 EVALUATING GENERALIZATION PERFORMANCE

We focus on evaluating the performance of our attention model by assessing its generalization error, measuring the ICL performance. For a given set of parameters $(\boldsymbol{V}, \boldsymbol{M})$, the model's generalization (ICL) error is defined as:

$$\mathcal{G}(\boldsymbol{V}, \boldsymbol{M}) := \mathbb{E}_{(\boldsymbol{Z}, y_l) \sim \mathcal{D}^{test}}\left[(y_l - \hat{y}(\boldsymbol{Z}; \boldsymbol{V}, \boldsymbol{M}))^2\right], \tag{8}$$

where $\mathcal{D}^{test}$ denotes the distribution of the test set, which includes input-output pairs generated with tasks that the model has not encountered during training. Since the task vectors in the test set differ from those encountered during training, the model is required to infer these new vectors based solely on the provided context. Therefore, the ICL/generalization error (8) assesses the genuine ICL capabilities of the model.

### 3.6 BAYES-OPTIMAL RIDGE ESTIMATOR

The Bayes-optimal ridge estimator provides a principled framework for estimating the task vector $\boldsymbol{w}$ given a prior distribution and a set of $l-1$ samples. It is defined as:

$$\hat{\boldsymbol{w}}_{Bayes} = \left( \frac{\bar{\boldsymbol{X}}^T \bar{\boldsymbol{X}}}{\sigma^2} + \boldsymbol{\Sigma}_w^{-1} \right)^{-1} \left( \frac{\bar{\boldsymbol{X}}^T \bar{\boldsymbol{y}}}{\sigma^2} + \boldsymbol{\Sigma}_w^{-1} \boldsymbol{\mu}_w \right), \tag{9}$$

where $\bar{\boldsymbol{X}}$ is the centered input matrix and $\bar{\boldsymbol{y}}$ is the centered label vector. This estimator combines information from observed data with prior knowledge of the distribution of $\boldsymbol{w}$, thereby balancing bias and variance. It serves as the gold standard against which we benchmark model predictions. The terms involving $\boldsymbol{\Sigma}_w^{-1}$ introduce a regularization effect, which is particularly advantageous in high-dimensional regimes.

The derivation, provided in Appendix A, illustrates how Bayesian principles inform regression by integrating data evidence with prior distributions to yield more reliable predictions. In our setting, the inputs and labels are derived from the prompt matrix $\boldsymbol{Z}$, and the Bayes-optimal linear model predicts any input $\boldsymbol{x}$ as $\hat{\boldsymbol{w}}_{Bayes}^T \boldsymbol{x}$.

## 4 THEORETICAL RESULTS

In this section, we present our main theoretical results on the ICL under distribution shifts for the Transformer with a linearized attention without MLP layers, denoted by (5). We begin by showing how to pretrain the model to approximate the Bayes-optimal linear predictor, thereby grounding its predictive performance. We then identify specific conditions under which the model fails to generalize under distribution shifts at test time, revealing key limitations of the model in ICL. Following this, we provide a detailed characterization of its generalization error, offering a principled framework for analyzing performance. Finally, we investigate the role of the temperature parameter and demonstrate that adjusting it appropriately can substantially improve generalization—especially in cases where the model initially fails to perform effective in-context learning.

### 4.1 MODEL PRETRAINING

We begin our pretraining analysis by observing that the prediction generated by the Transformer (5) can be reduced to the following form (see Appendix E for the derivation):

$$\hat{y}(\boldsymbol{Z}; \boldsymbol{V}, \boldsymbol{M}) := E_{d+1,l} = \frac{1}{\tau} \hat{\boldsymbol{w}}_{Att}(\boldsymbol{C}_{xx}, \boldsymbol{C}_{xy}, C_{yy}; \boldsymbol{M}, \boldsymbol{V})^T \boldsymbol{x}_l + b_{Att}(\boldsymbol{s}_x, s_y; \boldsymbol{V}), \tag{10}$$

where $\hat{\boldsymbol{w}}_{Att}(\boldsymbol{C}_{xx}, \boldsymbol{C}_{xy}, C_{yy}; \boldsymbol{M}, \boldsymbol{V}) \in \mathbb{R}^d$ and $b_{Att}(\boldsymbol{s}_x, s_y; \boldsymbol{V}) \in \mathbb{R}$. $\boldsymbol{s}_x$ and $s_y$ denote the sample means of the input $\boldsymbol{x}$ and the label $y$, respectively, and $\boldsymbol{C}_{xx}$ and $\boldsymbol{C}_{xy}$ are the sample covariances corresponding to $\mathrm{Cov}(\boldsymbol{x})$ and $\mathrm{Cov}(\boldsymbol{x}, y)$. These statistics are computed from the prompt matrix $\boldsymbol{Z}$.

For pretraining, we optimize the parameters $\boldsymbol{V}$ and $\boldsymbol{M}$ using $m$ samples of $(\boldsymbol{Z}, y_l)$ drawn from the distribution $\mathcal{D}^{train}$, where each $\boldsymbol{Z}$ contains $l-1$ $(\boldsymbol{x}, y)$ pairs intended for ICL. Building upon prior work that connects ICL in linear regression to the Bayes-optimal ridge estimator (Zhang et al., 2024; Raventós et al., 2023), we configure $\boldsymbol{M}$ and $\boldsymbol{V}$ to emulate Bayes-optimal ridge regression. Specifically, we aim for $\hat{\boldsymbol{w}}_{Att}(\boldsymbol{C}_{xx}, \boldsymbol{C}_{xy}; \boldsymbol{M}, \boldsymbol{V}) \approx \hat{\boldsymbol{w}}_{Bayes}$ and $b_{Att}(\boldsymbol{s}_x, s_y; \boldsymbol{V}) \approx 0$.

**Lemma 4.1** (Pretrained Parameters). *When the temperature parameter is set to $\tau = 1$ during pretraining, the following parameter configuration approximates the Bayes-optimal estimator in (9):*

$$\boldsymbol{M}_{11} = d \left( \frac{\hat{\boldsymbol{X}}^T \hat{\boldsymbol{X}}}{ml} + \frac{\sigma^2}{l} \boldsymbol{\Sigma}_w^{-1} \right)^{-1}, \qquad \boldsymbol{m}_{21} = \boldsymbol{0}, \tag{11}$$

$$\boldsymbol{v}_{21} = \frac{\sigma^2}{dl} \left( \frac{\hat{\boldsymbol{X}}^T \hat{\boldsymbol{X}}}{ml} \right)^{-1} \boldsymbol{\Sigma}_w^{-1} \boldsymbol{\mu}_w, \qquad v_{22} = \frac{1}{d},$$

*where $\hat{\boldsymbol{X}} \in \mathbb{R}^{ml \times d}$ is the centered input matrix formed from $ml$ samples of $\boldsymbol{x}$. This configuration aligns the our model with Bayes-optimal ridge regression. The quantities $\boldsymbol{\mu}_w$ and $\boldsymbol{\Sigma}_w$ can be estimated from the pretraining data. A detailed derivation is provided in Appendix F.*

This lemma establishes a theoretical connection between the pretrained parameters and the Bayes-optimal estimator, reinforcing the foundation of our approach.

Moreover, specific instances of Lemma 4.1 recover settings explored in prior studies. For example, under the assumptions $\boldsymbol{\Sigma}_x = \boldsymbol{\Sigma}_w = \boldsymbol{I}$, $\boldsymbol{\mu}_w = \boldsymbol{0}$, and $\sigma = 0$, Von Oswald et al. (2023) employ $\boldsymbol{M}_{11} = \text{Cov}(x)^{-1}$ and $\boldsymbol{v}_{21} = \boldsymbol{0}$ within a linear attention framework. Our formulation generalizes this by allowing $\boldsymbol{v}_{21} \neq \boldsymbol{0}$, which reflects our assumption that $\boldsymbol{\mu}_w \neq \boldsymbol{0}$—a departure from earlier works. Indeed, our analysis reveals that $\boldsymbol{v}_{21}$ encodes information related to task vector bias $\boldsymbol{\mu}_w$. Additionally, our choice of $\boldsymbol{M}_{11}$ explicitly accounts for label noise ($\sigma^2$), thereby enhancing the model's adaptability and maintaining a Bayesian interpretation.

We conclude this section with two remarks on task diversity and parameter optimality:

*Remark* 4.2. A high degree of task diversity (i.e., the number of distinct tasks) is essential for enabling effective in-context learning (Wu et al., 2024). Within our framework, task diversity directly impacts the accuracy of estimating $\boldsymbol{\mu}_w$ and $\boldsymbol{\Sigma}_w$ during pretraining.

*Remark* 4.3. While the pretrained parameters specified in Lemma 4.1 are not guaranteed to be optimal in all settings, they are analytically useful for examining the effects of distribution shifts and the role of the temperature parameter in ICL. Importantly, our characterization of ICL performance and temperature optimality does not depend on these particular parameter choices.

Based on Lemma 4.1, we arrive at the following corollary:

**Corollary 4.4.** *Suppose there is no distribution shift between training and inference. Then, under the parameter configuration of Lemma 4.1, the Transformer model (5) emulates the Bayes-optimal linear model, implying that it is capable of in-context learning according to Definition 3.3.*

Since the pretrained model succeeds in ICL for $\mathcal{D}^{test} = \mathcal{D}^{train}$, we next investigate how distribution shifts affect its ICL performance.

## 4.2 EFFECT OF DISTRIBUTION SHIFT

In this section, we explore scenarios where $\mathcal{D}^{test} \neq \mathcal{D}^{train}$, indicating a shift in the input, task, or noise distribution after pretraining the model. We consider three cases: (1) a shift in the input distribution (altered mean or covariance), (2) a shift in the task distribution, and (3) a change in the noise levels.

To evaluate the impact of these distribution shifts on ICL performance, we assess whether adjustments to $\boldsymbol{M}$ and/or $\boldsymbol{V}$ are necessary to match the Bayes-optimal linear model under the new distribution. If so, the model is considered sensitive to the shift. Otherwise, it is deemed robust.

**Case I: Shift in input distribution —** Recall that inputs are drawn as $\boldsymbol{x}_i \sim \mathcal{N}(\boldsymbol{\mu}_x, \boldsymbol{\Sigma}_x)$, as defined in (2). Let $\boldsymbol{\mu}_x^{train}, \boldsymbol{\Sigma}_x^{train}$ and $\boldsymbol{\mu}_x^{test}, \boldsymbol{\Sigma}_x^{test}$ denote the input means and covariances for pretraining and testing, respectively. We consider two subcases:

(i) Mean shift ($\boldsymbol{\mu}_x^{train} \neq \boldsymbol{\mu}_x^{test}$): Centering renders the linearized model invariant to mean shifts, but the uncentered linear attention model remains sensitive, as noted in Remark 3.4.

(ii) Covariance shift ($\boldsymbol{\Sigma}_x^{train} \neq \boldsymbol{\Sigma}_x^{test}$): Since $\boldsymbol{M}_{11}$ is fitted to the pretraining covariance, a mismatch drives the estimator away from Bayes-optimality, echoing prior results on linear attention (Zhang et al., 2024).

**Case II: Shift in Task Distribution —** The task vectors follow $\boldsymbol{w} \sim \mathcal{N}(\boldsymbol{\mu}_w, \boldsymbol{\Sigma}_w)$. Let $\boldsymbol{\mu}_w^{train}, \boldsymbol{\Sigma}_w^{train}$ and $\boldsymbol{\mu}_w^{test}, \boldsymbol{\Sigma}_w^{test}$ be the mean and covariance of the task distribution during pretraining and testing, respectively. The Transformer model can incorporate $\boldsymbol{\mu}_w^{train}$ and $\boldsymbol{\Sigma}_w^{train}$ via the pretrained parameters $\boldsymbol{M}_{11}$ and $\boldsymbol{v}_{21}$ (see Lemma 4.1). However, as the context length $l$ increases, the model's dependence on the task distribution diminishes. Thus, shifts in the task distribution primarily affect ICL performance for small $l$.

**Case III: Shift in noise distribution —** Finally, consider a change in the noise distribution: $\epsilon_i \sim \mathcal{N}(0, \sigma^2)$, with $\sigma_{train}^2$ and $\sigma_{test}^2$ denoting pretraining and testing noise variances. If $\sigma_{train}^2 \neq \sigma_{test}^2$, the parameters $\boldsymbol{M}_{11}$ and $\boldsymbol{v}_{21}$ become suboptimal relative to the Bayes-optimal linear model. However, as with the task distribution, the impact of noise shift diminishes as $l \to \infty$.

**Summary —** The Transformer model is robust to shifts in input mean but sensitive to input covariance changes. Shifts in task or noise distribution reduce ICL performance at small $l$, though

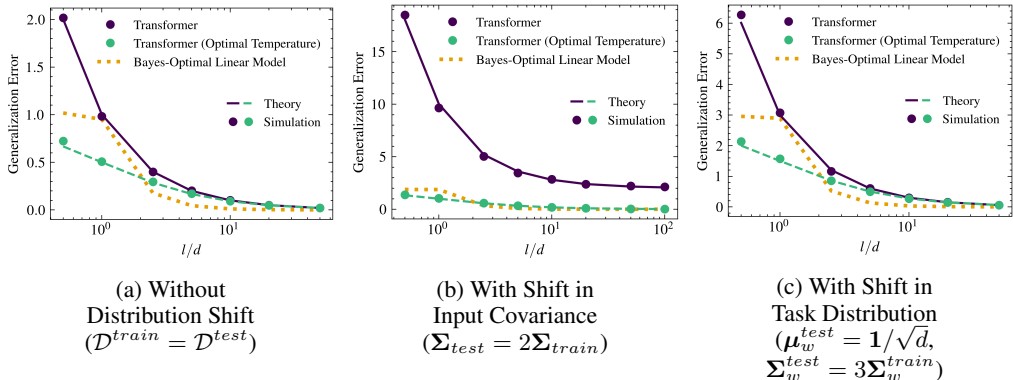

Figure 1: Experiments with Transformer (5) on ICL under distribution shifts. Parameters are set using (11) while the optimal temperature is calculated by Theorem 4.7. Here, $d = 50$, $m = 5000$ (with a new task per sample), $\sigma = 0.1$, $\boldsymbol{\mu}_x^{train} = \boldsymbol{\mu}_w^{train} = \mathbf{0}$, and $\boldsymbol{\Sigma}_x^{train} = \boldsymbol{\Sigma}_w^{train} = \boldsymbol{I}$.

increasing $l$ mitigates these effects. In Section 4.4, we explore optimal temperature selection as a way to enhance robustness. Before that, we analyze the generalization error of the model in the next section.

### 4.3 IN-CONTEXT LEARNING PERFORMANCE

We analyze the in-context learning (ICL) performance of the Transformer model (5) by evaluating the generalization error defined in (8). To establish a general setting for the subsequent results, we impose the following assumption on the pretrained parameters:

**Assumption 4.5.** There exists a constant $c > 0$ such that

$$\|\boldsymbol{M}_{11}\| \leq cd, \quad \|\boldsymbol{m}_{21}\| = 0, \quad \|\boldsymbol{v}_{21}\| \leq \frac{c}{dl}, \quad |v_{22}| \leq \frac{c}{d}.$$

Note that the pretrained parameters obtained in Lemma 4.1 satisfy Assumption 4.5 with high probability under Assumptions 3.1–3.2. However, the generalization error result stated below holds for any parameters $\boldsymbol{M}, \boldsymbol{V}$ that satisfy Assumption 4.5.

**Theorem 4.6** (Generalization error for ICL). *Suppose Assumptions 3.1–3.2 and 4.5 hold. At test time, assume the input, task, and noise distributions are given by $\mathcal{N}(\boldsymbol{\mu}_x, \boldsymbol{\Sigma}_x)$, $\mathcal{N}(\boldsymbol{\mu}_w, \boldsymbol{\Sigma}_w)$, and $\mathcal{N}(0, \sigma^2)$, respectively. Define*

$$\boldsymbol{A} := \boldsymbol{\Sigma}_x + \boldsymbol{\mu}_x \boldsymbol{\mu}_x^T, \quad \boldsymbol{B} := \boldsymbol{\Sigma}_w + \boldsymbol{\mu}_w \boldsymbol{\mu}_w^T.$$

*Then, the generalization error is*

$$\mathcal{G}(\boldsymbol{V}, \boldsymbol{M}) = \frac{1}{\tau^2} \operatorname{Tr}\left(\boldsymbol{A} \boldsymbol{M}_{11}^T \boldsymbol{F}_1 \boldsymbol{M}_{11}\right) - \frac{1}{\tau} \operatorname{Tr}\left(\boldsymbol{A}\left(\boldsymbol{F}_2 \boldsymbol{M}_{11} + \boldsymbol{M}_{11}^T \boldsymbol{F}_2^T\right)\right) + \operatorname{Tr}\left(\boldsymbol{A}\boldsymbol{B}\right) + \sigma^2, \quad (12)$$

*where*

$$\boldsymbol{F}_1 := \left(\boldsymbol{\Sigma}_x \hat{\boldsymbol{B}} + \frac{1}{l}\left(v_{22}^2 \sigma^2 + \operatorname{Tr}(\hat{\boldsymbol{B}} \boldsymbol{\Sigma}_x)\right) \boldsymbol{I}\right) \boldsymbol{\Sigma}_x, \quad \boldsymbol{F}_2 := (\boldsymbol{\mu}_w \boldsymbol{v}_{21}^T + v_{22} \boldsymbol{B}) \boldsymbol{\Sigma}_x, \quad (13)$$

$$\hat{\boldsymbol{B}} := v_{22} \boldsymbol{\mu}_w \boldsymbol{v}_{21}^T + v_{22} \boldsymbol{v}_{21} \boldsymbol{\mu}_w^T + v_{22}^2 \boldsymbol{B}. \quad (14)$$

*Proof.* The generalization error is derived using Isserlis' theorem (Isserlis, 1918) to compute higher-order moments. See Appendix G for the full derivation. $\square$

Theorem 4.6 illustrates how the parameters $\boldsymbol{M}$, $\boldsymbol{V}$, and the test-time data distribution affect the generalization error. Notably, the temperature parameter $\tau$ plays a critical role.

Although temperature can be implicitly encoded in $\boldsymbol{M}$ during pretraining, it becomes especially important under distribution shifts that the model is not equipped to handle. In such cases, one can optimize generalization performance by choosing the temperature $\tau_{\text{optimal}}$ that minimizes the generalization error, as discussed next.

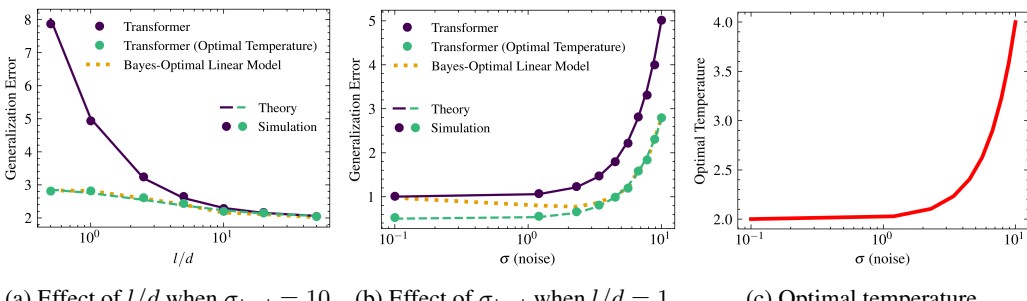

(a) Effect of $l/d$ when $\sigma_{test} = 10$    (b) Effect of $\sigma_{test}$ when $l/d = 1$    (c) Optimal temperature

Figure 2: Effect of noise shift on Transformer (5). The pretraining noise is $\sigma_{train} = 0.1$, while $\sigma_{test}$ varies across plots. The optimal temperature is set by Theorem 4.7. This setting matches Figure 1a, except for changes in test-time noise $\sigma_{test}$.

### 4.4 OPTIMAL ATTENTION TEMPERATURE IMPROVES PERFORMANCE

To address distribution shifts, we define the optimal attention temperature as follows:

**Theorem 4.7** (Optimal attention temperature). *Suppose Assumptions 3.1, 3.2, and 4.5 hold. To minimize the generalization error, the optimal attention temperature for inference is given by*

$$\tau_{optimal} = \frac{2Tr\left(\boldsymbol{A}\boldsymbol{M}_{11}^T\boldsymbol{F}_1\boldsymbol{M}_{11}\right)}{Tr\left(\boldsymbol{A}\left(\boldsymbol{F}_2\boldsymbol{M}_{11} + \boldsymbol{M}_{11}^T\boldsymbol{F}_2^T\right)\right)}, \tag{15}$$

*provided that $Tr\left(\boldsymbol{A}\left(\boldsymbol{F}_2\boldsymbol{M}_{11} + \boldsymbol{M}_{11}^T\boldsymbol{F}_2^T\right)\right) > 0$ and $Tr\left(\boldsymbol{A}\boldsymbol{M}_{11}^T\boldsymbol{F}_1\boldsymbol{M}_{11}\right) > 0$.*

*Proof.* We minimize the generalization error from Theorem 4.6 with respect to $\tau$ (Appendix I). □

Consider the optimal temperature $\tau_{\text{optimal}}$ from Theorem 4.7. When $\tau_{\text{optimal}} \neq 1$, using an unadjusted temperature leads to suboptimal generalization error. Thus, incorporating the optimal temperature improves generalization in in-context learning under distribution shift.

A natural question is whether the optimal temperature can completely mitigate the adverse effects of distribution shifts. This depends on both the pretraining and test distributions. In some settings, the adjustment can entirely compensate for the shift. For example, if the task distribution is fixed as $\boldsymbol{w} \sim \mathcal{N}(\boldsymbol{0}, \boldsymbol{I})$, the noise variance is $\sigma = 0$, and the input distribution changes from $\boldsymbol{x}_{\text{train}} \sim \mathcal{N}(\boldsymbol{0}, \boldsymbol{I})$ to $\boldsymbol{x}_{\text{test}} \sim \mathcal{N}(\boldsymbol{0}, c\boldsymbol{I})$, then the optimal temperature $\tau_{\text{optimal}} = c$ fully counteracts the shift. In more complex scenarios, it may only partially mitigate the impact, yet still yields improved ICL. See Appendix M for an extended discussion of the closed-form optimal temperature (15).

## 5 EXPERIMENTAL RESULTS

In this section, we empirically validate our theory and show that optimal attention temperature consistently enhances generalization. We begin with controlled linear regression experiments using (i) the simplified Transformer model with linearized attention (5) and (ii) GPT-2 (Radford et al., 2019), which combines multi-head softmax attention with MLP layers[2]. These experiments confirm that our theoretical insights transfer from simplified to expressive architectures. Finally, we evaluate Llama2-7B (Touvron et al., 2023) on SCIQ in-context learning tasks (Welbl et al., 2017), demonstrating that temperature selection is a principled and effective lever for improving robustness in large language models.

### 5.1 EXPERIMENTS ON LINEAR REGRESSION TASKS

We consider a Transformer architecture with linearized attention and no MLP layers, as analyzed in our theoretical development. Figures 1 and 2 illustrate its behavior on linear regression tasks (2). Theoretical predictions closely match empirical performance across a range of conditions, confirming the robustness of our analysis. In Figure 1, we compare the ICL performance of the model with and without applying the optimal temperature. As context length $l$ increases (Figure 1a), the model's

---

[2]GPT-2 results are in Appendix K.

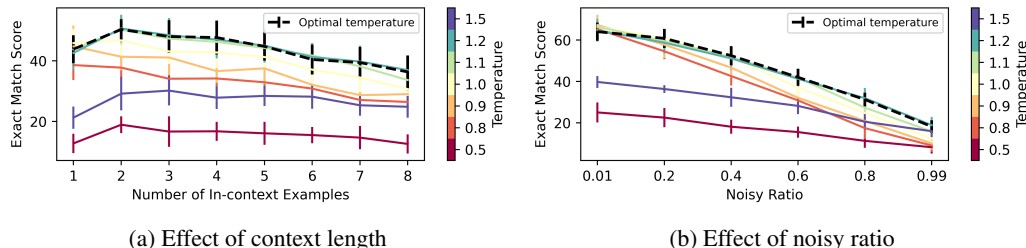

(a) Effect of context length

(b) Effect of noisy ratio

Figure 3: Effect of attention temperature on the ICL performance of LLaMA-2-7B (Touvron et al., 2023) on the SCIQ dataset (Welbl et al., 2017). Distribution shift is induced by injecting noisy yet "relevant" labels into in-context demonstrations following Gao et al. (2024). Panel (a) fixes the noisy ratio at 0.6; panel (b) fixes the number of in-context examples at 6. Results (averaged over 12 Monte Carlo runs) include error bars showing one standard deviation. Attention temperature of all the layers is set to $\tau\sqrt{d_k}$ for dimension independence, where $d_k$ denotes the key dimension of the corresponding layer. Furthermore, the dashed black line marks the "optimal temperature" computed from the variance-to-mean ratio of pre-softmax scores, which is an insight derived from Theorem 4.7, as explained in Appendix J. Full experimental details appear in Appendix K.

predictions converge to those of the Bayes-optimal linear model, validating its ICL capability. Figure 1b shows that under an input covariance shift, model performance degrades—but applying the optimal temperature restores alignment with the Bayes-optimal solution. Additionally, Figure 1c shows that the influence of task distribution shift decreases as $l$ increases.

We further evaluate robustness to label noise in Figure 2. In Figure 2a, we observe that noise effects diminish as the context length increases, consistent with our theoretical predictions. However, at small $l$, temperature adjustment becomes critical. In Figure 2b (for $l = d$), the Transformer increasingly diverges from the Bayes-optimal model as noise grows, yet optimal temperature correction closes this gap. Figure 2c shows that the optimal temperature increases with noise level, indicating a principled relationship between noise and temperature under limited context.

## 5.2 Experiments with LLMs for in-context question answering tasks

To assess the practical relevance of our theoretical framework, we investigate how attention temperature impacts the ICL behavior of LLMs. Since the optimal temperature in Theorem 4.7 is not directly applicable here due to setting differences, we derive insights regarding temperature choice in other settings based on the optimal temperature in Theorem 4.7. Specifically, the insight is that the temperature choice should be proportional to the ratio of the variance of pre-softmax scores to the mean of those, which is described in Appendix J in detail.

Following Gao et al. (2024), we generate SCIQ-based (Welbl et al., 2017) ICL tasks that introduce distribution shift via noisy labels, with prompt examples and label construction detailed in Appendix K. We evaluate Llama2-7B (Touvron et al., 2023) using exact-match score.

Figure 3 shows the results. In (a) and (b), optimal attention temperature enhances the ICL performance for various context lengths and noisy ratios (analogous to Figures 2a and 2b). Furthermore, in (b), higher noise ratios push the optimal temperature upward, matching our theoretical prediction (cf. Figure 2c). Together, these experiments demonstrate that the optimal temperature is not only theoretically motivated but also an effective tool for improving ICL robustness in real-world LLMs.

## 6 Conclusion

This work provides a unified theoretical and empirical account of how attention temperature governs the in-context learning (ICL) performance of pretrained Transformers under distribution shift. Using a simplified yet expressive framework based on *linearized softmax attention*, we analytically show how shifts in input covariance and label noise degrade ICL and derive an *optimal temperature* that provably minimizes generalization error. Extensive experiments on synthetic regression tasks, GPT-2, and LLaMA-2 validate our predictions, demonstrating that temperature selection is not a mere heuristic but a principled mechanism for improving robustness. Taken together, our results advance the theoretical understanding of Transformer behavior under distribution shift and establish attention temperature as a powerful, practical lever for building more adaptive and generalizable foundation models.

## ETHICS STATEMENT

Our research, as presented in the paper, conforms with the Code of Ethics. As the work is mostly of a theoretical nature, it does not involve any component that can be subject to ethical concerns.

## REPRODUCIBILITY STATEMENT

All results and settings are explained in detail to streamline reproducibility. Derivations of the theoretical results can be found in the appendix. The settings used to generate the figures are explained in Section 3, in the corresponding captions, and in the appendix. Finally, the code for the experimental results will be released with the camera-ready version of this work.

## THE USE OF LARGE LANGUAGE MODELS (LLMS)

During the writing of the paper, we utilize LLMs in order to sharpen the presentation language. In doing so, we provide a sentence or a paragraph that we wrote before, instruct an LLM model to rewrite it in a better tone, and use the produced sentence/paragraph whenever it clearly describes what we aim to describe. This approach has been repeatedly applied to multiple parts of the paper to refine the writing. Overall, we take full responsibility for the contents written.

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

# A    DERIVATION OF BAYES-OPTIMAL RIDGE ESTIMATOR FOR $w$

We derive the Bayes-optimal ridge estimator for $w$ given a set of context samples. We place a Gaussian prior on $w$, assumed to be a random vector $w \sim \mathcal{N}(\mu_0, \Sigma_0)$ with prior mean $\mu_0$ and covariance $\Sigma_0$. Let the observed (centered) inputs and labels be

$$\bar{X} = [\bar{x}_1, \ldots, \bar{x}_{l-1}]^T, \quad \bar{y} = [\bar{y}_1, \ldots, \bar{y}_{l-1}]^T,$$

and assume i.i.d. Gaussian noise $\epsilon_i \sim \mathcal{N}(0, \sigma^2)$. The likelihood of $\bar{y}$ given $w$ is

$$p(\bar{y} \mid \bar{X}, w) = \prod_{i=1}^{l-1} \frac{1}{\sqrt{2\pi\sigma^2}} \exp\left[-\frac{(\bar{y}_i - w^T \bar{x}_i)^2}{2\sigma^2}\right] \tag{16}$$

$$\propto \exp\left[-\frac{1}{2\sigma^2}(\bar{y} - \bar{X}w)^T(\bar{y} - \bar{X}w)\right], \tag{17}$$

where $\propto$ denotes proportionality.

By Bayes' rule, the posterior of $w$ is proportional to the product of likelihood and prior:

$$p(w \mid \bar{y}, \bar{X}) \propto p(\bar{y} \mid \bar{X}, w)\, p(w). \tag{18}$$

Substituting the Gaussian prior yields

$$p(w \mid \bar{y}, \bar{X}) \propto \exp\left[-\frac{1}{2\sigma^2}(\bar{y} - \bar{X}w)^T(\bar{y} - \bar{X}w)\right] \exp\left[-\frac{1}{2}(w - \mu_0)^T \Sigma_0^{-1}(w - \mu_0)\right]. \tag{19}$$

To determine the form of the posterior distribution, we complete the square in the exponent by collecting all terms involving $w$. Expanding the exponent in the joint expression from above, we obtain:

$$-\frac{1}{2\sigma^2}\left(\bar{y}^T\bar{y} - 2\bar{y}^T\bar{X}w + w^T\bar{X}^T\bar{X}w\right) - \frac{1}{2}\left(w^T\Sigma_0^{-1}w - 2\mu_0^T\Sigma_0^{-1}w + \mu_0^T\Sigma_0^{-1}\mu_0\right). \tag{20}$$

Grouping the quadratic and linear terms in $w$, we arrive at:

$$-\frac{1}{2}w^T\left(\frac{\bar{X}^T\bar{X}}{\sigma^2} + \Sigma_0^{-1}\right)w + w^T\left(\frac{\bar{X}^T\bar{y}}{\sigma^2} + \Sigma_0^{-1}\mu_0\right) + \text{terms independent of } w. \tag{21}$$

Defining the posterior precision and linear coefficient terms as $\Sigma_l^{-1} = \frac{\bar{X}^T\bar{X}}{\sigma^2} + \Sigma_0^{-1}$ and $b_l = \frac{\bar{X}^T\bar{y}}{\sigma^2} + \Sigma_0^{-1}\mu_0$, the exponent can be rewritten as

$$-\frac{1}{2}w^T\Sigma_l^{-1}w + w^Tb_l = -\frac{1}{2}(w - \mu_l)^T\Sigma_l^{-1}(w - \mu_l) + \text{const}, \tag{22}$$

where $\mu_l = \Sigma_l b_l$ denotes the posterior mean. Expanding this expression gives:

$$\mu_l = \left(\frac{\bar{X}^T\bar{X}}{\sigma^2} + \Sigma_0^{-1}\right)^{-1}\left(\frac{\bar{X}^T\bar{y}}{\sigma^2} + \Sigma_0^{-1}\mu_0\right). \tag{23}$$

Hence, the posterior distribution of $w$ given the observed data is Gaussian:

$$w \mid \bar{y}, \bar{X} \sim \mathcal{N}(\mu_l, \Sigma_l), \tag{24}$$

where $\mu_l$ is the posterior mean and $\Sigma_l$ is the posterior covariance matrix.

Under squared-error loss, the Bayes-optimal estimator coincides with the posterior mean, yielding the Bayes-optimal ridge estimator:

$$\hat{w}_{\text{Ridge}} = \mathbb{E}[w \mid \bar{y}, \bar{X}] = \mu_l = \left(\frac{\bar{X}^T\bar{X}}{\sigma^2} + \Sigma_0^{-1}\right)^{-1}\left(\frac{\bar{X}^T\bar{y}}{\sigma^2} + \Sigma_0^{-1}\mu_0\right). \tag{25}$$

This expression provides the Bayes-optimal ridge estimate of $w$ under a Gaussian prior and additive Gaussian noise—minimizing expected squared error with respect to the posterior.

# B DERIVATION OF LINEARIZED SOFTMAX

The function softmax : $\mathbb{R}^l \to \mathbb{R}^l$ is defined component-wise as

$$\text{softmax}(\boldsymbol{z})_i := \frac{e^{z_i}}{\sum_{j=1}^{l} e^{z_j}} \quad \forall i \in \{1, \dots, l\}. \tag{26}$$

To obtain a linear approximation, we expand around the origin $\boldsymbol{z} = \boldsymbol{0}$ using a first-order Taylor series:

$$\text{softmax}(\boldsymbol{z}) \approx \text{softmax}(\boldsymbol{0}) + J_{\text{softmax}}(\boldsymbol{0})\boldsymbol{z}, \tag{27}$$

where $J_{\text{softmax}}(\boldsymbol{0})$ is the Jacobian matrix of the softmax function evaluated at $\boldsymbol{z} = \boldsymbol{0}$.

We first compute the zeroth-order term:

$$\text{softmax}(\boldsymbol{0}) = \frac{e^0}{\sum_{j=1}^{l} e^0}\boldsymbol{1} = \frac{1}{l}\boldsymbol{1}. \tag{28}$$

Next, we evaluate the Jacobian entries at $\boldsymbol{z} = \boldsymbol{0}$:

$$J_{\text{softmax}}(\boldsymbol{0})_{ii} = \text{softmax}(\boldsymbol{0})_i \left(1 - \text{softmax}(\boldsymbol{0})_i\right) = \frac{l-1}{l^2}, \quad \forall i, \tag{29}$$

$$J_{\text{softmax}}(\boldsymbol{0})_{ij} = -\text{softmax}(\boldsymbol{0})_i \cdot \text{softmax}(\boldsymbol{0})_j = -\frac{1}{l^2}, \quad \forall i \neq j. \tag{30}$$

This yields the compact matrix form:

$$J_{\text{softmax}}(\boldsymbol{0}) = \frac{1}{l}\boldsymbol{I} - \frac{1}{l^2}\boldsymbol{1}\boldsymbol{1}^T. \tag{31}$$

Substituting back, we obtain the linearized softmax:

$$\text{softmax}(\boldsymbol{z}) \approx \frac{1}{l}\boldsymbol{1} + \left(\frac{1}{l}\boldsymbol{I} - \frac{1}{l^2}\boldsymbol{1}\boldsymbol{1}^T\right)\boldsymbol{z}, \tag{32}$$

$$= \left(\frac{1}{l} - \frac{1}{l^2}\sum_{j=1}^{l} z_j\right)\boldsymbol{1} + \frac{1}{l}\boldsymbol{z}, \tag{33}$$

$$=: \text{linearized\_softmax}(\boldsymbol{z}). \tag{34}$$

This derivation yields the linearized attention formulation in (5). From a practical standpoint, linearized attention mechanisms have been empirically evaluated and shown to achieve performance comparable to standard softmax attention (Han et al., 2024).

# C LINEAR VS. LINEARIZED ATTENTION FOR IN-CONTEXT LEARNING

Here, we highlight the distinction between linear attention and linearized attention in the context of the linear regression problem defined in (2). Analytically, the key difference lies in the fact that the linearized attention model operates on centered input data, whereas the linear attention model uses raw data without centering. Apart from this centering step, both mechanisms are equivalent, except that linearized attention includes an additional bias term ($b_{Att}$ in (10)). However, this bias term is inconsequential in our linear regression setting and does not affect the predictive outcome.

Thus, the data-centering operation is the principal differentiator in our analysis. Specifically, linear attention's omission of centering makes it sensitive to shifts in the input mean, whereas linearized attention remains robust under such transformations. We illustrate this phenomenon in Figure 4, where we simulate a shift in the input mean at test time. The results demonstrate that linear attention fails to recover Bayes-optimal performance under mean shift, indicating its limitations for in-context learning in this setting. In contrast, linearized attention successfully compensates for the mean shift and achieves Bayes-optimal performance as the number of context points $l$ increases.

Therefore, in the presence of possible distributional shifts—particularly in the input mean—linearized attention offers a more robust and theoretically grounded alternative to linear attention for in-context learning.

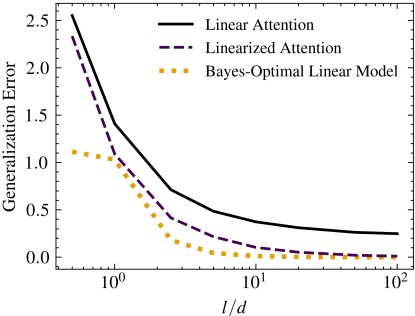

Figure 4: Comparison of linear and linearized attention under a shift in input mean. The plot illustrates the impact of a test-time shift in input mean on the performance of linear attention and linearized attention. While linear attention degrades under the distribution shift and fails to recover the Bayes-optimal performance, linearized attention remains robust and asymptotically matches the Bayes-optimal predictor as the number of context length $l$ increases.

## D  TEMPERATURE EFFECTS FOR SOFTMAX AND LINEARIZED SOFTMAX

The temperature parameter in softmax directly controls the variance of the output distribution. At higher temperatures, the variance across components decreases, and in the limit $\tau \to \infty$, all elements converge to $1/l$ with zero variance. Conversely, lower temperatures increase variance, and as $\tau \to 0^+$, the output approaches a one-hot vector, achieving maximal variance.

In the linearized case, temperature similarly acts as an inverse scaling of the variance of the output components, capturing the limit $\tau \to \infty$ (all elements equal to $1/l$). For $\tau \to 0^+$, linearized softmax also reflects the maximal variance, but it does not produce a true one-hot distribution. Thus, linearized softmax closely mirrors the temperature behavior of softmax, except in the degenerate limit $\tau \to 0^+$, which is not of practical relevance in this work.

To further illustrate these effects, Figure 5 compares softmax and linearized softmax across different temperatures. The figure demonstrates that linearized softmax faithfully captures the variance effect of temperature: the variance of the output components is inversely proportional to $\tau$. Moreover, as $\tau \to \infty$, both softmax and linearized softmax concentrate around $1/l$, whereas linear attention with temperature scaling does not. Overall, the output distributions of softmax and linearized softmax are highly similar, except at very small values of $\tau$, where linearized softmax may yield negative components while softmax tends toward sparsity with many zeros. By contrast, linear attention with temperature scaling produces qualitatively different distributions. This comparison highlights the advantage of linearized softmax as a faithful surrogate for analyzing temperature effects relevant to softmax.

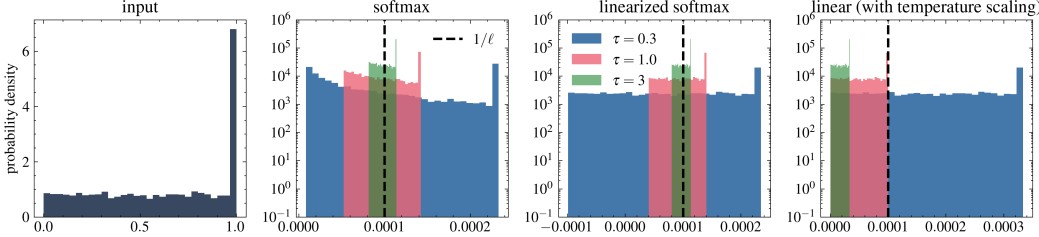

Figure 5: Comparison of temperature effects of softmax, linearized softmax, and linear (with temperature scaling) cases. We consider an input vector $\boldsymbol{x} \in \mathbb{R}^l$ whose histogram is illustrated on the left-most plot. Rest of the plots illustrates histograms of the elements of $\text{softmax}(\boldsymbol{x}/\tau)$, $\text{linearized\_softmax}(\boldsymbol{x}/\tau)$ defined in (34) and $\boldsymbol{x}/(l\tau)$ from left to right, respectively.

## E   EXPANDED FORM OF LINEARIZED ATTENTION

Using block matrix notation, the prediction from the linearized attention model can be expanded as:

$$\hat{y}(\boldsymbol{Z}; \boldsymbol{V}, \boldsymbol{M}) = A_{d+1,l}, \tag{35}$$

$$= \frac{1}{l}\boldsymbol{V}_{d+1,:}\boldsymbol{Z}\left(\frac{(\boldsymbol{KZ})^T(\boldsymbol{QZ}_{:,l})}{\tau} - \frac{\mathbf{1}}{l}\sum_{j=1}^{l}\frac{(\boldsymbol{KZ}_{:,j})^T(\boldsymbol{QZ}_{:,l})}{\tau} + \mathbf{1}\right), \tag{36}$$

$$= \frac{1}{l}[\boldsymbol{v}_{21}^T \ v_{22}]\boldsymbol{Z}\left(\frac{\boldsymbol{Z}^T\boldsymbol{MZ}_{:,l}}{\tau} - \frac{\mathbf{1}}{l}\sum_{j=1}^{l}\frac{(\boldsymbol{Z}_{:,j})^T\boldsymbol{MZ}_{:,l}}{\tau} + \mathbf{1}\right), \tag{37}$$

$$= \frac{1}{l}[\boldsymbol{v}_{21}^T \ v_{22}][\boldsymbol{X} \ \boldsymbol{y}]^T\left(\frac{[\boldsymbol{X} \ \boldsymbol{y}]\boldsymbol{M}[\boldsymbol{x}_l^T \ 0]^T}{\tau} - \frac{\mathbf{1}}{l}\sum_{j=1}^{l}\frac{[\boldsymbol{x}_i^T \ y_i]\boldsymbol{M}[\boldsymbol{x}_l^T \ 0]^T}{\tau} + \mathbf{1}\right), \tag{38}$$

$$= \frac{1}{l}[\boldsymbol{v}_{21}^T \ v_{22}][\boldsymbol{X} \ \boldsymbol{y}]^T\left(\frac{1}{\tau}\left[\boldsymbol{X} - \mathbf{1}\boldsymbol{s}_x^T \ \boldsymbol{y} - s_y\mathbf{1}\right]\begin{bmatrix}\boldsymbol{M}_{11} & * \\ \boldsymbol{m}_{21}^T & *\end{bmatrix}[\boldsymbol{x}_l^T \ 0]^T + \mathbf{1}\right), \tag{39}$$

$$= \frac{1}{l}[\boldsymbol{v}_{21}^T \ v_{22}][\boldsymbol{X} \ \boldsymbol{y}]^T\left(\frac{1}{\tau}\left(\boldsymbol{X} - \mathbf{1}\boldsymbol{s}_x^T\right)\boldsymbol{M}_{11}\boldsymbol{x}_l + \frac{1}{\tau}\left(\boldsymbol{y} - s_y\mathbf{1}\right)\boldsymbol{m}_{21}^T\boldsymbol{x}_l + \mathbf{1}\right), \tag{40}$$

$$= \frac{1}{l}\left(\boldsymbol{v}_{21}^T\boldsymbol{X}^T + v_{22}\boldsymbol{y}^T\right)\left(\frac{1}{\tau}\left(\boldsymbol{X} - \mathbf{1}\boldsymbol{s}_x^T\right)\boldsymbol{M}_{11}\boldsymbol{x}_l + \frac{1}{\tau}\left(\boldsymbol{y} - s_y\mathbf{1}\right)\boldsymbol{m}_{21}^T\boldsymbol{x}_l + \mathbf{1}\right), \tag{41}$$

$$= \frac{1}{\tau}\left(\boldsymbol{v}_{21}^T\left(\frac{\boldsymbol{X}^T\boldsymbol{X}}{l} - \boldsymbol{s}_x\boldsymbol{s}_x^T\right) + v_{22}\left(\frac{\boldsymbol{y}^T\boldsymbol{X}}{l} - s_y\boldsymbol{s}_x^T\right)\right)\boldsymbol{M}_{11}\boldsymbol{x}_l,$$

$$\quad + \frac{1}{\tau}\left(\boldsymbol{v}_{21}^T\left(\frac{\boldsymbol{X}^T\boldsymbol{y}}{l} - s_y\boldsymbol{s}_x\right) + v_{22}\left(\frac{\boldsymbol{y}^T\boldsymbol{y}}{l} - s_y^2\right)\right)\boldsymbol{m}_{21}^T\boldsymbol{x}_l + \boldsymbol{v}_{21}^T\boldsymbol{s}_x + v_{22}s_y, \tag{42}$$

$$= \frac{1}{\tau}\left(\boldsymbol{v}_{21}^T\boldsymbol{C}_{xx} + v_{22}\boldsymbol{C}_{xy}^T\right)\boldsymbol{M}_{11}\boldsymbol{x}_l + \frac{1}{\tau}\left(\boldsymbol{v}_{21}^T\boldsymbol{C}_{xy} + v_{22}C_{yy}\right)\boldsymbol{m}_{21}^T\boldsymbol{x}_l + \boldsymbol{v}_{21}^T\boldsymbol{s}_x + v_{22}s_y, \tag{43}$$

$$= \frac{1}{\tau}\left(\left(\boldsymbol{v}_{21}^T\boldsymbol{C}_{xx} + v_{22}\boldsymbol{C}_{xy}^T\right)\boldsymbol{M}_{11} + \left(\boldsymbol{v}_{21}^T\boldsymbol{C}_{xy} + v_{22}C_{yy}\right)\boldsymbol{m}_{21}^T\right)\boldsymbol{x}_l + \boldsymbol{v}_{21}^T\boldsymbol{s}_x + v_{22}s_y, \tag{44}$$

where the summary statistics are defined as:

$$\boldsymbol{s}_x := \frac{1}{l}\sum_{i=1}^{l}\boldsymbol{x}_i, \qquad\qquad s_y := \frac{1}{l}\sum_{i=1}^{l-1}y_i,$$

$$\boldsymbol{C}_{xx} := \frac{1}{l}\sum_{i=1}^{l}\boldsymbol{x}_i\boldsymbol{x}_i^T - \boldsymbol{s}_x\boldsymbol{s}_x^T, \qquad \boldsymbol{C}_{xy} := \frac{1}{l}\sum_{i=1}^{l-1}y_i\boldsymbol{x}_i - s_y\boldsymbol{s}_x, \qquad C_{yy} := \frac{1}{l}\sum_{i=1}^{l-1}y_i^2 - s_y^2.$$

Then, we define

$$\hat{\boldsymbol{w}}_{Att}(\boldsymbol{C}_{xx}, \boldsymbol{C}_{xy}, C_{yy}; \boldsymbol{M}, \boldsymbol{V}) = \boldsymbol{M}_{11}^T\left(\boldsymbol{C}_{xx}\boldsymbol{v}_{21} + v_{22}\boldsymbol{C}_{xy}\right) + \left(\boldsymbol{v}_{21}^T\boldsymbol{C}_{xy} + v_{22}C_{yy}\right)\boldsymbol{m}_{21}, \tag{45}$$

$$b_{Att}(\boldsymbol{s}_x, s_y; \boldsymbol{V}) = \boldsymbol{v}_{21}^T\boldsymbol{s}_x + v_{22}s_y, \tag{46}$$

which allows us to write

$$\hat{y}(\boldsymbol{Z}; \boldsymbol{V}, \boldsymbol{M}) = \frac{1}{\tau}\hat{\boldsymbol{w}}_{Att}(\boldsymbol{C}_{xx}, \boldsymbol{C}_{xy}, C_{yy}; \boldsymbol{M}, \boldsymbol{V})^T\boldsymbol{x}_l + b_{Att}(\boldsymbol{s}_x, s_y; \boldsymbol{V}). \tag{47}$$

## F   DERIVATION OF THE PRETRAINING FOR ICL BY MIMICKING THE BAYES-OPTIMAL ESTIMATOR

Here, we derive the pretraining of the linearized attention model by mimicking the Bayes-optimal ridge estimator (9). Recall that the prediction of the linearized attention model is

$$\hat{y}(\boldsymbol{Z}; \boldsymbol{V}, \boldsymbol{M}) = \frac{1}{\tau}\hat{\boldsymbol{w}}_{Att}(\boldsymbol{C}_{xx}, \boldsymbol{C}_{xy}, C_{yy}; \boldsymbol{M}, \boldsymbol{V})^T\boldsymbol{x}_l + b_{Att}(\boldsymbol{s}_x, s_y; \boldsymbol{V}), \tag{48}$$

which is derived in Appendix E. Furthermore, the Bayes-optimal ridge regression model's prediction is

$$\hat{y}_{Bayes} = \hat{\boldsymbol{w}}_{Bayes}^T \boldsymbol{x}_l. \tag{49}$$

Therefore, we select the parameters $\boldsymbol{M}$ and $\boldsymbol{V}$ such that

$$\hat{\boldsymbol{w}}_{Att}(\boldsymbol{C}_{xx}, \boldsymbol{C}_{xy}, C_{yy}; \boldsymbol{M}, \boldsymbol{V}) \approx \hat{\boldsymbol{w}}_{Bayes}, \quad b_{Att}(\boldsymbol{s}_x, s_y; \boldsymbol{V}) \approx 0, \tag{50}$$

which makes the prediction of the linearized attention model approximately equal to that of the Bayes-optimal regression. Furthermore, we consider $\tau = 1$ for the pretraining. Let's first focus on $\hat{\boldsymbol{w}}_{Att}(\boldsymbol{C}_{xx}, \boldsymbol{C}_{xy}, C_{yy}; \boldsymbol{M}, \boldsymbol{V})$ as follows

$$\hat{\boldsymbol{w}}_{Att}(\boldsymbol{C}_{xx}, \boldsymbol{C}_{xy}, C_{yy}; \boldsymbol{M}, \boldsymbol{V})$$
$$= \left( \boldsymbol{M}_{11}^T \left( \boldsymbol{C}_{xx} \boldsymbol{v}_{21} + v_{22} \boldsymbol{C}_{xy} \right) + \left( \boldsymbol{v}_{21}^T \boldsymbol{C}_{xy} + v_{22} C_{yy} \right) \boldsymbol{m}_{21} \right), \tag{51}$$
$$= \left( \boldsymbol{M}_{11}^T \left( \frac{\bar{\boldsymbol{X}}^T \bar{\boldsymbol{X}}}{l} \boldsymbol{v}_{21} + v_{22} \frac{\bar{\boldsymbol{X}}^T \bar{\boldsymbol{y}}}{l} \right) + \left( \boldsymbol{v}_{21}^T \frac{\bar{\boldsymbol{X}}^T \bar{\boldsymbol{y}}}{l} + v_{22} \frac{\bar{\boldsymbol{y}}^T \bar{\boldsymbol{y}}}{l} \right) \boldsymbol{m}_{21} \right). \tag{52}$$

To reach the last line, we use the fact that $\boldsymbol{C}_{xx} := \boldsymbol{X}^T \boldsymbol{X}/l - \boldsymbol{s}_x \boldsymbol{s}_x^T = \bar{\boldsymbol{X}}^T \bar{\boldsymbol{X}}/l, \boldsymbol{C}_{xy} := \boldsymbol{X}^T \boldsymbol{y}/l - s_y \boldsymbol{s}_x = \bar{\boldsymbol{X}}^T \bar{\boldsymbol{y}}/l$ and $C_{yy} = \bar{\boldsymbol{y}}^T \bar{\boldsymbol{y}}/l$, where $\bar{\boldsymbol{X}} := \boldsymbol{X} - \boldsymbol{s}_x^T$ and $\bar{\boldsymbol{y}} := \boldsymbol{y} - s_y$ denote centered input matrix and centered label vector. Now, recall that the Bayes-optimal ridge estimator is

$$\hat{\boldsymbol{w}}_{Bayes} = \left( \frac{\bar{\boldsymbol{X}}^T \bar{\boldsymbol{X}}}{\sigma^2} + \boldsymbol{\Sigma}_w^{-1} \right)^{-1} \left( \frac{\bar{\boldsymbol{X}}^T \bar{\boldsymbol{y}}}{\sigma^2} + \boldsymbol{\Sigma}_w^{-1} \boldsymbol{\mu}_w \right), \tag{53}$$

as derived in Appendix A. Looking at equations (53) and (52) together, we can see that setting the parameters as follows would make $\hat{\boldsymbol{w}}_{Att} = \hat{\boldsymbol{w}}_{Bayes}$ hold

$$\boldsymbol{M}_{11} = \frac{l}{\sigma^2} \left( \frac{\bar{\boldsymbol{X}}^T \bar{\boldsymbol{X}}}{\sigma^2} + \boldsymbol{\Sigma}_w^{-1} \right)^{-1}, \quad \boldsymbol{v}_{21} = \frac{\sigma^2}{l} \left( \frac{\bar{\boldsymbol{X}}^T \bar{\boldsymbol{X}}}{l} \right)^{-1} \boldsymbol{\Sigma}_w^{-1} \boldsymbol{\mu}_w, \quad \boldsymbol{m}_{21} = \boldsymbol{0}, \quad v_{22} = 1. \tag{54}$$

However, while Bayes-optimal estimator $\hat{\boldsymbol{w}}_{Bayes}$ is different for each sample, the attention model should be pretrained and fixed. Thus, we replace $\bar{\boldsymbol{X}}^T \bar{\boldsymbol{X}}$ in (54) with $\hat{\boldsymbol{X}}^T \hat{\boldsymbol{X}}/m$ as follows, where $\hat{\boldsymbol{X}} \in \mathbb{R}^{ml \times d}$ is the centred input matrix including all the (pre)training data consisting of $ml$ samples.

$$\boldsymbol{M}_{11} = \frac{l}{\sigma^2} \left( \frac{\hat{\boldsymbol{X}}^T \hat{\boldsymbol{X}}}{m\sigma^2} + \boldsymbol{\Sigma}_w^{-1} \right)^{-1}, \quad \boldsymbol{v}_{21} = \frac{\sigma^2}{l} \left( \frac{\hat{\boldsymbol{X}}^T \hat{\boldsymbol{X}}}{ml} \right)^{-1} \boldsymbol{\Sigma}_w^{-1} \boldsymbol{\mu}_w, \quad \boldsymbol{m}_{21} = \boldsymbol{0}, \quad v_{22} = 1. \tag{55}$$

In practice, the variance of noise $\sigma^2$, the mean $\boldsymbol{\mu}_w$, and covariance $\boldsymbol{\Sigma}_w$ of the task vectors are unknown. Yet, we can use their estimates based on the (pre)training data.

Now, we can focus on making $b_{Att}(\boldsymbol{s}_x, s_y; \boldsymbol{V}) \approx 0$ hold as follows

$$b_{Att}(\boldsymbol{s}_x, s_y; \boldsymbol{V}) = \boldsymbol{v}_{21}^T \boldsymbol{s}_x + v_{22} s_y, \tag{56}$$

where $\boldsymbol{s}_x$ and $s_y$ are based on data so we have no control over them. Instead, by using Assumptions 3.1 and 3.2, we can choose $\boldsymbol{v}_{21}$ and $v_{22}$ such that $b_{Att} \to 0$ as $l, d \to \infty$. Note that Assumption 3.1 makes $\boldsymbol{v}_{21}^T \boldsymbol{s}_x + v_{22} s_y$ bounded with high probability for $\boldsymbol{v}_{21}$ and $v_{22}$ given in (55). Therefore, multiplying $\boldsymbol{v}_{21}, v_{22}$ given in (55) with $1/d$ would make $b_{Att} \to 0$ as $d \to \infty$. To fix the impact of the multiplication for $\hat{\boldsymbol{w}}_{Att}$, we can multiply $\boldsymbol{M}_{11}$ with $d$ as well. So, by applying the mentioned multiplications, we reach the following pretrained parameters mimicking the Bayes-optimal regression model

$$\boldsymbol{M}_{11} = \frac{dl}{\sigma^2} \left( \frac{\hat{\boldsymbol{X}}^T \hat{\boldsymbol{X}}}{m\sigma^2} + \boldsymbol{\Sigma}_w^{-1} \right)^{-1}, \quad \boldsymbol{v}_{21} = \frac{\sigma^2}{dl} \left( \frac{\hat{\boldsymbol{X}}^T \hat{\boldsymbol{X}}}{ml} \right)^{-1} \boldsymbol{\Sigma}_w^{-1} \boldsymbol{\mu}_w, \quad \boldsymbol{m}_{21} = \boldsymbol{0}, \quad v_{22} = \frac{1}{d}. \tag{57}$$

## G  CHARACTERIZATION OF GENERALIZATION ERROR FOR ICL UNDER DISTRIBUTION SHIFT

Here, we characterize the generalization error for in-context learning under distribution shift, given that $M$ and $V$ are pretrained and fixed. So, the impact of pretraining distribution $\mathcal{D}^{train}$ is captured by $M$ and $V$. Suppose that $\mathcal{D}^{test}$ denotes the test distribution. To avoid additional notations, here, we again use $\boldsymbol{\mu}_x, \boldsymbol{\mu}_w, \boldsymbol{\Sigma}_x, \boldsymbol{\Sigma}_w, \sigma^2$ to denote means and covariances for input and task vectors and noise variance for the inference (test). However, note that these can be different from those used for pretraining. We begin studying the generalization error defined in (8) as follows

$$\mathcal{G}(\boldsymbol{V}, \boldsymbol{M}) := \mathbb{E}_{(\boldsymbol{Z}, y_l) \sim \mathcal{D}^{test}} \left[ (y_l - \hat{y}(\boldsymbol{Z}; \boldsymbol{V}, \boldsymbol{M}))^2 \right], \tag{58}$$

$$= \mathbb{E}_{(\boldsymbol{Z}, y_l) \sim \mathcal{D}^{test}} \left[ \left( \frac{1}{\tau} \hat{\boldsymbol{w}}_{Att}(\boldsymbol{C}_{xx}, \boldsymbol{C}_{xy}, C_{yy}; \boldsymbol{M}, \boldsymbol{V})^T \boldsymbol{x}_l + b_{Att}(\boldsymbol{s}_x, s_y; \boldsymbol{V}) - y_l \right)^2 \right], \tag{59}$$

$$= \mathbb{E}_{(\boldsymbol{Z}, y_l) \sim \mathcal{D}^{test}} \left[ \left( \frac{1}{\tau} \left( \boldsymbol{M}_{11}^T \left( \boldsymbol{C}_{xx} \boldsymbol{v}_{21} + v_{22} \boldsymbol{C}_{xy} \right) \right)^T \boldsymbol{x}_l - y_l \right)^2 \right], \tag{60}$$

where we use the parameters from pretraining (57) together with Assumptions 3.1 and 3.2 to reach the last line. Then,

$$\mathcal{G}(\boldsymbol{V}, \boldsymbol{M}) = \mathbb{E}_{(\boldsymbol{Z}, y_l) \sim \mathcal{D}^{test}} \left[ \left( \frac{1}{\tau} \left( \boldsymbol{M}_{11}^T \left( \boldsymbol{C}_{xx} \boldsymbol{v}_{21} + v_{22} \boldsymbol{C}_{xy} \right) \right)^T \boldsymbol{x}_l - y_l \right)^2 \right], \tag{61}$$

$$= \mathbb{E} \left[ \left( \frac{1}{\tau} \left( \boldsymbol{M}_{11}^T \left( \frac{1}{l} \sum_{i \leq l} \bar{\boldsymbol{x}}_i \bar{\boldsymbol{x}}_i^T \boldsymbol{v}_{21} + v_{22} \frac{1}{l} \sum_{i \leq l-1} \bar{\boldsymbol{x}}_i (\bar{\boldsymbol{x}}_i^T \boldsymbol{w} + \epsilon_i) \right) \right)^T \boldsymbol{x}_l - \boldsymbol{w}^T \boldsymbol{x}_l - \epsilon_l \right)^2 \right], \tag{62}$$

$$= \mathbb{E} \left[ \left( \frac{1}{\tau} \left( \boldsymbol{M}_{11}^T \left( \frac{1}{l} \sum_{i \leq l} \bar{\boldsymbol{x}}_i \bar{\boldsymbol{x}}_i^T \boldsymbol{v}_{21} + v_{22} \frac{1}{l} \sum_{i \leq l-1} \bar{\boldsymbol{x}}_i (\bar{\boldsymbol{x}}_i^T \boldsymbol{w} + \epsilon_i) \right) \right)^T \boldsymbol{x}_l - \boldsymbol{w}^T \boldsymbol{x}_l \right)^2 \right] + \sigma^2 \tag{63}$$

where $\bar{\boldsymbol{x}}_i := \boldsymbol{x}_i - \boldsymbol{s}_x = \boldsymbol{x}_i - \frac{1}{l} \sum_{i \leq l} \boldsymbol{x}_i$ and we use $\epsilon_l \sim \mathcal{N}(0, \sigma^2)$ to reach the final line. We continue by defining

$$\boldsymbol{w}_{diff} := \frac{1}{\tau} \boldsymbol{M}_{11}^T \left( \frac{1}{l} \sum_{i \leq l-1} \bar{\boldsymbol{x}}_i \bar{\boldsymbol{x}}_i^T \boldsymbol{v}_{21} + v_{22} \frac{1}{l} \sum_{i \leq l-1} \bar{\boldsymbol{x}}_i (\bar{\boldsymbol{x}}_i^T \boldsymbol{w} + \epsilon_i) \right) - \boldsymbol{w}, \tag{64}$$

which allows us to write

$$\mathcal{G}(\boldsymbol{V}, \boldsymbol{M}) = \mathbb{E} \left[ \left( \boldsymbol{w}_{diff}^T \boldsymbol{x}_l \right)^2 \right] + \sigma^2, \tag{65}$$

$$= \mathbb{E} \left[ \boldsymbol{w}_{diff}^T \mathbb{E}_{\boldsymbol{x}_l} [\boldsymbol{x}_l \boldsymbol{x}_l^T] \boldsymbol{w}_{diff} \right] + \sigma^2, \tag{66}$$

$$= \mathbb{E} \left[ \boldsymbol{w}_{diff}^T \left( \boldsymbol{\mu}_x \boldsymbol{\mu}_x^T + \boldsymbol{\Sigma}_x \right) \boldsymbol{w}_{diff} \right] + \sigma^2, \tag{67}$$

by the law of total expectation since $\boldsymbol{w}_{diff}$ is independent of $\boldsymbol{x}_l$. Note that when writing (65), we safely ignore terms with $(1/l) \bar{\boldsymbol{x}}_l \bar{\boldsymbol{x}}_l^T \boldsymbol{v}_{21}$ in (63) since they vanish as $l \to \infty$ by Assumptions 3.1-3.2 and 4.5. Letting $\boldsymbol{A} := \boldsymbol{\mu}_x \boldsymbol{\mu}_x^T + \boldsymbol{\Sigma}_x$, we write

$$\mathcal{G}(\boldsymbol{V}, \boldsymbol{M}) = \mathbb{E} \left[ \boldsymbol{w}_{diff}^T \boldsymbol{A} \boldsymbol{w}_{diff} \right] + \sigma^2, \tag{68}$$

$$= \mathbb{E} \left[ \text{Tr} \left( \boldsymbol{w}_{diff}^T \boldsymbol{A} \boldsymbol{w}_{diff} \right) \right] + \sigma^2, \tag{69}$$

$$= \mathbb{E} \left[ \text{Tr} \left( \boldsymbol{A} \boldsymbol{w}_{diff} \boldsymbol{w}_{diff}^T \right) \right] + \sigma^2, \tag{70}$$

$$= \text{Tr} \left( \boldsymbol{A} \mathbb{E}[\boldsymbol{w}_{diff} \boldsymbol{w}_{diff}^T] \right) + \sigma^2, \tag{71}$$

where we first apply the cyclic property of trace and then use the linearity of expectation and trace to reach the last line. Now, we need to calculate $\mathbb{E}[\boldsymbol{w}_{diff}\boldsymbol{w}_{diff}^T]$, for which we first take the expectation over $\boldsymbol{w}$. To do so, we rewrite $\boldsymbol{w}_{diff}$ as

$$\boldsymbol{w}_{diff} = \underbrace{\frac{1}{\tau}\boldsymbol{M}_{11}^T \left(\frac{1}{l}\sum_{i\leq l-1}\bar{\boldsymbol{x}}_i\bar{\boldsymbol{x}}_i^T\boldsymbol{v}_{21} + v_{22}\frac{1}{l}\sum_{i\leq l-1}\bar{\boldsymbol{x}}_i\epsilon_i\right)}_{\boldsymbol{e}} + \underbrace{\left(\frac{v_{22}}{\tau}\boldsymbol{M}_{11}^T\frac{1}{l}\sum_{i\leq l-1}\bar{\boldsymbol{x}}_i\bar{\boldsymbol{x}}_i^T - \boldsymbol{I}\right)}_{\boldsymbol{D}}\boldsymbol{w},$$

(72)

$$= \boldsymbol{e} + \boldsymbol{D}\boldsymbol{w},$$

(73)

where we define

$$\boldsymbol{e} := \frac{1}{\tau}\boldsymbol{M}_{11}^T\left(\frac{1}{l}\sum_{i\leq l-1}\bar{\boldsymbol{x}}_i\bar{\boldsymbol{x}}_i^T\boldsymbol{v}_{21} + v_{22}\frac{1}{l}\sum_{i\leq l-1}\bar{\boldsymbol{x}}_i\epsilon_i\right),$$

(74)

$$\boldsymbol{D} := \left(\frac{v_{22}}{\tau}\boldsymbol{M}_{11}^T\frac{1}{l}\sum_{i\leq l-1}\bar{\boldsymbol{x}}_i\bar{\boldsymbol{x}}_i^T - \boldsymbol{I}\right).$$

(75)

Since $\boldsymbol{e}$ and $\boldsymbol{D}$ are independent of $\boldsymbol{w}$, we can easily calculate $\mathbb{E}_{\boldsymbol{w}}[\boldsymbol{w}_{diff}\boldsymbol{w}_{diff}^T]$ as follows

$$\mathbb{E}\left[\mathbb{E}_{\boldsymbol{w}}[\boldsymbol{w}_{diff}\boldsymbol{w}_{diff}^T]\right] = \mathbb{E}\left[\mathbb{E}_{\boldsymbol{w}}[(\boldsymbol{e} + \boldsymbol{D}\boldsymbol{w})(\boldsymbol{e} + \boldsymbol{D}\boldsymbol{w})^T]\right],$$

(76)

$$= \mathbb{E}\left[\boldsymbol{e}\boldsymbol{e}^T\right] + \mathbb{E}\left[\boldsymbol{e}\boldsymbol{\mu}_w^T\boldsymbol{D}^T\right] + \mathbb{E}\left[\boldsymbol{D}\boldsymbol{\mu}_w\boldsymbol{e}^T\right] + \mathbb{E}\left[\boldsymbol{D}(\boldsymbol{\mu}_x\boldsymbol{\mu}_x^T + \boldsymbol{\Sigma}_w)\boldsymbol{D}^T\right],$$

(77)

$$= \mathbb{E}\left[\boldsymbol{e}\boldsymbol{e}^T\right] + \mathbb{E}\left[\boldsymbol{D}\boldsymbol{\mu}_w\boldsymbol{e}^T\right]^T + \mathbb{E}\left[\boldsymbol{D}\boldsymbol{\mu}_w\boldsymbol{e}^T\right] + \mathbb{E}\left[\boldsymbol{D}\boldsymbol{B}\boldsymbol{D}^T\right],$$

(78)

where we first apply the law of total expectation, then take the expectation over $\boldsymbol{w}$ and finally, we define $\boldsymbol{B} := \boldsymbol{\mu}_x\boldsymbol{\mu}_x^T + \boldsymbol{\Sigma}_w$ to reach the last line. Note that $\boldsymbol{\mu}_w$ and $\boldsymbol{B}$ are fixed while $\boldsymbol{e}$ and $\boldsymbol{D}$ are random in the last line. Therefore, we are required to calculate the three expectations that appeared in (78).

Before getting into the calculations of the aforementioned expectations, we provide the following lemma that is useful for the calculation of the expectations.

**Lemma G.1.** *Let $\bar{\boldsymbol{x}} \sim \mathcal{N}(\boldsymbol{0}, \boldsymbol{\Sigma})$, where $\bar{\boldsymbol{x}} \in \mathbb{R}^d$. Let $\bar{\boldsymbol{x}}_i$ be $l-1$ independent samples of $\bar{\boldsymbol{x}}$ for $i = 1, \ldots, l-1$. Furthermore, let $\boldsymbol{A}$ be a fixed $d \times d$ matrix. Then, the following holds*

$$\mathbb{E}\left[\left(\frac{1}{l}\sum_{i\leq l-1}\bar{\boldsymbol{x}}_i\bar{\boldsymbol{x}}_i^T\right)\boldsymbol{A}\left(\frac{1}{l}\sum_{i\leq l-1}\bar{\boldsymbol{x}}_i\bar{\boldsymbol{x}}_i^T\right)\right] = \frac{l-1}{l}\boldsymbol{\Sigma}\boldsymbol{A}\boldsymbol{\Sigma} + \frac{1}{l}\boldsymbol{\Sigma}\boldsymbol{A}^T\boldsymbol{\Sigma} + \frac{1}{l}Tr(\boldsymbol{A}\boldsymbol{\Sigma})\boldsymbol{\Sigma}. \quad (79)$$

*Proof.* This is proven by using Isserlis' theorem (Isserlis, 1918) in Appendix H. $\qquad\square$

Note that our inputs $\bar{\boldsymbol{x}}_i$ are centered, i.e., $\bar{\boldsymbol{x}}_i = \boldsymbol{x}_i - \frac{1}{l}\sum_{i\leq l}\boldsymbol{x}_i$, so their distribution is $\mathcal{N}(\boldsymbol{0}, \boldsymbol{\Sigma}_x)$ as $l \to \infty$. Therefore, Lemma G.1 is directly applicable in our setting.

Next, we start the calculations of the expectations in (78) with $\mathbb{E}\left[ee^T\right]$ as follows

$$\mathbb{E}\left[ee^T\right] = \frac{1}{\tau^2}M_{11}^T\mathbb{E}\left[\left(\frac{1}{l}\sum_{i\le l-1}\bar{x}_i\bar{x}_i^Tv_{21} + v_{22}\frac{1}{l}\sum_{i\le l-1}\bar{x}_i\epsilon_i\right)\right.$$
$$\left.\cdot\left(\frac{1}{l}\sum_{i\le l-1}v_{21}^T\bar{x}_i\bar{x}_i^T + v_{22}\frac{1}{l}\sum_{i\le l-1}\bar{x}_i^T\epsilon_i\right)\right]M_{11}, \tag{80}$$

$$= \frac{1}{\tau^2}M_{11}^T\left(\mathbb{E}\left[\left(\frac{1}{l}\sum_{i\le l-1}\bar{x}_i\bar{x}_i^Tv_{21}\right)\left(\frac{1}{l}\sum_{i\le l-1}v_{21}^T\bar{x}_i\bar{x}_i^T\right)\right.\right.$$
$$\left.\left.+\left(v_{22}\frac{1}{l}\sum_{i\le l-1}\bar{x}_i\epsilon_i\right)\left(v_{22}\frac{1}{l}\sum_{i\le l-1}\bar{x}_i^T\epsilon_i\right)\right]\right)M_{11}, \tag{81}$$

$$= \frac{1}{\tau^2}M_{11}^T\left(\mathbb{E}\left[\left(\frac{1}{l}\sum_{i\le l-1}\bar{x}_i\bar{x}_i^T\right)v_{21}v_{21}^T\left(\frac{1}{l}\sum_{i\le l-1}\bar{x}_i\bar{x}_i^T\right)+\left(v_{22}^2\frac{\sigma^2}{l^2}\sum_{i\le l-1}\bar{x}_i\bar{x}_i^T\right)\right]\right)M_{11}, \tag{82}$$

$$= \frac{1}{\tau^2}M_{11}^T\left(\Sigma_x v_{21}v_{21}^T\Sigma_x + \frac{1}{l}\text{Tr}\left(v_{21}v_{21}^T\Sigma_x\right)\Sigma_x + v_{22}^2\frac{\sigma^2(l-1)}{l^2}\Sigma_x\right)M_{11}, \tag{83}$$

$$= \frac{1}{\tau^2}M_{11}^T\left(v_{22}^2\frac{\sigma^2}{l}\Sigma_x\right)M_{11}, \tag{84}$$

where we first use the independence of the random variables and $\epsilon_i \sim \mathcal{N}(0,\sigma^2)$ to simplify the equation. Then, we apply Lemma G.1 and use the fact that $\mathbb{E}[\bar{x}_i\bar{x}_i^T] = \Sigma_x$ to get the penultimate line. Finally, we drop the vanishing terms and simplify the result using Assumptions 3.1-3.2 and 4.5 in order to reach the last line.

We continue with the calculation of $\mathbb{E}\left[D\mu_w e^T\right]$ as

$$\mathbb{E}\left[D\mu_w e^T\right]$$
$$= \frac{1}{\tau}\mathbb{E}\left[\left(\frac{v_{22}}{\tau}M_{11}^T\frac{1}{l}\sum_{i\le l-1}\bar{x}_i\bar{x}_i^T - I\right)\mu_w\left(\frac{1}{l}\sum_{i\le l-1}v_{21}^T\bar{x}_i\bar{x}_i^T + v_{22}\frac{1}{l}\sum_{i\le l-1}\bar{x}_i^T\epsilon_i\right)\right]M_{11}, \tag{85}$$

$$= \frac{1}{\tau}\mathbb{E}\left[\left(\frac{v_{22}}{\tau}M_{11}^T\frac{1}{l}\sum_{i\le l-1}\bar{x}_i\bar{x}_i^T - I\right)\mu_w\left(\frac{1}{l}\sum_{i\le l-1}v_{21}^T\bar{x}_i\bar{x}_i^T\right)\right]M_{11}, \tag{86}$$

$$= \frac{1}{\tau}\frac{v_{22}}{\tau}M_{11}^T\mathbb{E}\left[\left(\frac{1}{l}\sum_{i\le l-1}\bar{x}_i\bar{x}_i^T\right)\mu_w v_{21}^T\left(\frac{1}{l}\sum_{i\le l-1}\bar{x}_i\bar{x}_i^T\right)\right]M_{11}$$
$$-\mu_w v_{21}^T\mathbb{E}\left[\frac{1}{l}\sum_{i\le l-1}\bar{x}_i\bar{x}_i^T\right]M_{11}, \tag{87}$$

$$= \frac{1}{\tau}\frac{v_{22}}{\tau}M_{11}^T\left(\Sigma_x\mu_w v_{21}^T\Sigma_x + \frac{1}{l}\Sigma_x v_{21}\mu_w^T\Sigma_x + \frac{1}{l}\text{Tr}\left(\mu_w v_{21}^T\Sigma_x\right)\Sigma_x\right)M_{11}$$
$$-\frac{1}{\tau}\frac{l-1}{l}\mu_w v_{21}^T\Sigma_x M_{11}, \tag{88}$$

$$= \frac{v_{22}}{\tau^2}M_{11}^T\left(\Sigma_x\mu_w v_{21}^T\Sigma_x + \frac{1}{l}\text{Tr}\left(\mu_w v_{21}^T\Sigma_x\right)\Sigma_x\right)M_{11} - \frac{1}{\tau}\mu_w v_{21}^T\Sigma_x M_{11}, \tag{89}$$

where we again first use the independence of the random variables and $\epsilon_i \sim \mathcal{N}(0,\sigma^2)$. Then, we apply basic algebraic manipulations. To reach the penultimate line, we utilize Lemma G.1 together with the fact that $\mathbb{E}[\bar{x}_i\bar{x}_i^T] = \Sigma_x$. Using Assumptions 3.1-3.2 and 4.5, we reach the last line.

Finally, we calculate $\mathbb{E}\left[\boldsymbol{DBD}^T\right]$ as follows

$$\mathbb{E}\left[\boldsymbol{DBD}^T\right]$$

$$= \mathbb{E}\left[\left(\frac{v_{22}}{\tau}\boldsymbol{M}_{11}^T \frac{1}{l}\sum_{i\leq l-1}\bar{\boldsymbol{x}}_i\bar{\boldsymbol{x}}_i^T - \boldsymbol{I}\right)\boldsymbol{B}\left(\frac{v_{22}}{\tau}\frac{1}{l}\sum_{i\leq l-1}\bar{\boldsymbol{x}}_i\bar{\boldsymbol{x}}_i^T \boldsymbol{M}_{11} - \boldsymbol{I}\right)\right], \quad (90)$$

$$= \mathbb{E}\left[\left(\frac{v_{22}}{\tau}\boldsymbol{M}_{11}^T \frac{1}{l}\sum_{i\leq l-1}\bar{\boldsymbol{x}}_i\bar{\boldsymbol{x}}_i^T\right)\boldsymbol{B}\left(\frac{v_{22}}{\tau}\frac{1}{l}\sum_{i\leq l-1}\bar{\boldsymbol{x}}_i\bar{\boldsymbol{x}}_i^T \boldsymbol{M}_{11}\right)\right]$$

$$- \mathbb{E}\left[\left(\frac{v_{22}}{\tau}\boldsymbol{M}_{11}^T \frac{1}{l}\sum_{i\leq l-1}\bar{\boldsymbol{x}}_i\bar{\boldsymbol{x}}_i^T\right)\boldsymbol{B}\right] - \mathbb{E}\left[\boldsymbol{B}\left(\frac{v_{22}}{\tau}\frac{1}{l}\sum_{i\leq l-1}\bar{\boldsymbol{x}}_i\bar{\boldsymbol{x}}_i^T \boldsymbol{M}_{11}\right)\right] + \boldsymbol{B}, \quad (91)$$

$$= \frac{v_{22}^2}{\tau^2}\boldsymbol{M}_{11}^T\mathbb{E}\left[\left(\frac{1}{l}\sum_{i\leq l-1}\bar{\boldsymbol{x}}_i\bar{\boldsymbol{x}}_i^T\right)\boldsymbol{B}\left(\frac{1}{l}\sum_{i\leq l-1}\bar{\boldsymbol{x}}_i\bar{\boldsymbol{x}}_i^T\right)\right]\boldsymbol{M}_{11}$$

$$- \frac{v_{22}}{\tau}\boldsymbol{M}_{11}^T\mathbb{E}\left[\left(\frac{1}{l}\sum_{i\leq l-1}\bar{\boldsymbol{x}}_i\bar{\boldsymbol{x}}_i^T\right)\right]\boldsymbol{B} - \frac{v_{22}}{\tau}\boldsymbol{B}\mathbb{E}\left[\left(\frac{1}{l}\sum_{i\leq l-1}\bar{\boldsymbol{x}}_i\bar{\boldsymbol{x}}_i^T\right)\right]\boldsymbol{M}_{11} + \boldsymbol{B}, \quad (92)$$

$$= \frac{v_{22}^2}{\tau^2}\boldsymbol{M}_{11}^T\left(\boldsymbol{\Sigma}_x\boldsymbol{B}\boldsymbol{\Sigma}_x + \frac{1}{l}\text{Tr}\left(\boldsymbol{B}\boldsymbol{\Sigma}_x\right)\boldsymbol{\Sigma}_x\right)\boldsymbol{M}_{11} - \frac{v_{22}}{\tau}\frac{l-1}{l}\boldsymbol{M}_{11}^T\boldsymbol{\Sigma}_x\boldsymbol{B}$$

$$- \frac{v_{22}}{\tau}\frac{l-1}{l}\boldsymbol{B}\boldsymbol{\Sigma}_x\boldsymbol{M}_{11} + \boldsymbol{B}, \quad (93)$$

$$= \frac{v_{22}^2}{\tau^2}\boldsymbol{M}_{11}^T\left(\boldsymbol{\Sigma}_x\boldsymbol{B}\boldsymbol{\Sigma}_x + \frac{1}{l}\text{Tr}\left(\boldsymbol{B}\boldsymbol{\Sigma}_x\right)\boldsymbol{\Sigma}_x\right)\boldsymbol{M}_{11} - \frac{v_{22}}{\tau}\boldsymbol{M}_{11}^T\boldsymbol{\Sigma}_x\boldsymbol{B} - \frac{v_{22}}{\tau}\boldsymbol{B}\boldsymbol{\Sigma}_x\boldsymbol{M}_{11} + \boldsymbol{B}, \quad (94)$$

where we first do basic algebraic manipulations. Then, we use Lemma G.1 and $\mathbb{E}[\bar{\boldsymbol{x}}_i\bar{\boldsymbol{x}}_i^T] = \boldsymbol{\Sigma}_x$ to get the penultimate line. For the final line, we utilize $l \to \infty$ by Assumption 3.2.

Putting the found expectation results into (78), we get

$$\mathbb{E}\left[\mathbb{E}_{\boldsymbol{w}}[\boldsymbol{w}_{diff}\boldsymbol{w}_{diff}^T]\right] = \mathbb{E}\left[\boldsymbol{e}\boldsymbol{e}^T\right] + \mathbb{E}\left[\boldsymbol{D}\boldsymbol{\mu}_w\boldsymbol{e}^T\right]^T + \mathbb{E}\left[\boldsymbol{D}\boldsymbol{\mu}_w\boldsymbol{e}^T\right] + \mathbb{E}\left[\boldsymbol{DBD}^T\right], \quad (95)$$

$$= \frac{1}{\tau^2}\boldsymbol{M}_{11}^T\boldsymbol{F}_1\boldsymbol{M}_{11} - \frac{1}{\tau}\boldsymbol{F}_2\boldsymbol{M}_{11} + \frac{1}{\tau}\boldsymbol{M}_{11}^T\boldsymbol{F}_2^T + \boldsymbol{B}. \quad (96)$$

where matrices $\boldsymbol{F}_1$ and $\boldsymbol{F}_2$ are defined as

$$\boldsymbol{F}_1 := v_{22}^2\frac{\sigma^2}{l}\boldsymbol{\Sigma}_x + v_{22}\left(\boldsymbol{\Sigma}_x\boldsymbol{\mu}_w\boldsymbol{v}_{21}^T\boldsymbol{\Sigma}_x + \frac{1}{l}\text{Tr}\left(\boldsymbol{\mu}_w\boldsymbol{v}_{21}^T\boldsymbol{\Sigma}_x\right)\boldsymbol{\Sigma}_x\right) \quad (97)$$

$$+ v_{22}\left(\boldsymbol{\Sigma}_x\boldsymbol{\mu}_w\boldsymbol{v}_{21}^T\boldsymbol{\Sigma}_x + \frac{1}{l}\text{Tr}\left(\boldsymbol{\mu}_w\boldsymbol{v}_{21}^T\boldsymbol{\Sigma}_x\right)\boldsymbol{\Sigma}_x\right)^T + v_{22}^2\left(\boldsymbol{\Sigma}_x\boldsymbol{B}\boldsymbol{\Sigma}_x + \frac{1}{l}\text{Tr}\left(\boldsymbol{B}\boldsymbol{\Sigma}_x\right)\boldsymbol{\Sigma}_x\right), \quad (98)$$

$$= \left(\boldsymbol{\Sigma}_x\hat{\boldsymbol{B}} + \left(v_{22}^2\frac{\sigma^2}{l} + \frac{1}{l}\text{Tr}\left(\hat{\boldsymbol{B}}\boldsymbol{\Sigma}_x\right)\right)\boldsymbol{I}\right)\boldsymbol{\Sigma}_x, \quad (98)$$

$$\boldsymbol{F}_2 := \boldsymbol{\mu}_w\boldsymbol{v}_{21}^T\boldsymbol{\Sigma}_x + v_{22}\boldsymbol{B}\boldsymbol{\Sigma}_x = (\boldsymbol{\mu}_w\boldsymbol{v}_{21}^T + v_{22}\boldsymbol{B})\boldsymbol{\Sigma}_x, \quad (99)$$

with $\hat{\boldsymbol{B}} := v_{22}\boldsymbol{\mu}_w\boldsymbol{v}_{21}^T + v_{22}\boldsymbol{v}_{21}\boldsymbol{\mu}_w^T + v_{22}^2\boldsymbol{B}$.

Going back to generalization error in (71), we have

$$\mathcal{G}(\boldsymbol{V}, \boldsymbol{M}) = \text{Tr}\left(\boldsymbol{A}\mathbb{E}[\boldsymbol{w}_{diff}\boldsymbol{w}_{diff}^T]\right) + \sigma^2, \quad (100)$$

$$= \text{Tr}\left(\boldsymbol{A}\left(\frac{1}{\tau^2}\boldsymbol{M}_{11}^T\boldsymbol{F}_1\boldsymbol{M}_{11} - \frac{1}{\tau}\boldsymbol{F}_2\boldsymbol{M}_{11} + \frac{1}{\tau}\boldsymbol{M}_{11}^T\boldsymbol{F}_2^T + \boldsymbol{B}\right)\right) + \sigma^2, \quad (101)$$

where $\boldsymbol{F}_1 = \left(\boldsymbol{\Sigma}_x\hat{\boldsymbol{B}} + \frac{1}{l}\left(v_{22}^2\sigma^2 + \text{Tr}\left(\hat{\boldsymbol{B}}\boldsymbol{\Sigma}_x\right)\right)\boldsymbol{I}\right)\boldsymbol{\Sigma}_x$, and $\boldsymbol{F}_2 = (\boldsymbol{\mu}_w\boldsymbol{v}_{21}^T + v_{22}\boldsymbol{B})\boldsymbol{\Sigma}_x$. Furthermore, $\hat{\boldsymbol{B}}$ is defined as $\hat{\boldsymbol{B}} := v_{22}\boldsymbol{\mu}_w\boldsymbol{v}_{21}^T + v_{22}\boldsymbol{v}_{21}\boldsymbol{\mu}_w^T + v_{22}^2\boldsymbol{B}$.

## H    PROOF OF LEMMA G.1

We first restate the lemma as follows.

Let $\bar{\boldsymbol{x}} \sim \mathcal{N}(0, \boldsymbol{\Sigma})$, where $\bar{\boldsymbol{x}} \in \mathbb{R}^d$. Let $\bar{\boldsymbol{x}}_i$ be $l$ independent samples of $\bar{\boldsymbol{x}}$ for $i = 1, \dots, l$. Let $\boldsymbol{A}$ be a fixed $d \times d$ matrix. Then, the following holds

$$\mathbb{E}\left[\left(\frac{1}{l}\sum_{i \leq l}\bar{\boldsymbol{x}}_i\bar{\boldsymbol{x}}_i^T\right)\boldsymbol{A}\left(\frac{1}{l}\sum_{i \leq l}\bar{\boldsymbol{x}}_i\bar{\boldsymbol{x}}_i^T\right)\right] = \boldsymbol{\Sigma}\boldsymbol{A}\boldsymbol{\Sigma} + \frac{1}{l}\boldsymbol{\Sigma}\boldsymbol{A}^T\boldsymbol{\Sigma} + \frac{1}{l}\mathrm{Tr}(\boldsymbol{A}\boldsymbol{\Sigma})\boldsymbol{\Sigma}. \tag{102}$$

*Proof.* Let $\boldsymbol{S}_x = \frac{1}{l}\sum_{i=1}^{l}\bar{\boldsymbol{x}}_i\bar{\boldsymbol{x}}_i^T$. First, note that $E[\bar{\boldsymbol{x}}_i\bar{\boldsymbol{x}}_i^T] = \boldsymbol{\Sigma}$ since $\bar{\boldsymbol{x}}_i \sim \mathcal{N}(0, \boldsymbol{\Sigma})$.

Thus, $\mathbb{E}[\boldsymbol{S}_x] = \frac{1}{l}\sum_{i=1}^{l}\mathbb{E}[\bar{\boldsymbol{x}}_i\bar{\boldsymbol{x}}_i^T] = \frac{1}{l}\sum_{i=1}^{l}\boldsymbol{\Sigma} = \boldsymbol{\Sigma}$. We have

$$\boldsymbol{S}_x\boldsymbol{A}\boldsymbol{S}_x = \frac{1}{l^2}\sum_{i=1}^{l}\sum_{j=1}^{l}\bar{\boldsymbol{x}}_i\bar{\boldsymbol{x}}_i^T\boldsymbol{A}\bar{\boldsymbol{x}}_j\bar{\boldsymbol{x}}_j^T \tag{103}$$

Taking the expectation, we get

$$\mathbb{E}[\boldsymbol{S}_x\boldsymbol{A}\boldsymbol{S}_x] = \frac{1}{l^2}\sum_{i=1}^{l}\sum_{j=1}^{l}\mathbb{E}[\bar{\boldsymbol{x}}_i\bar{\boldsymbol{x}}_i^T\boldsymbol{A}\bar{\boldsymbol{x}}_j\bar{\boldsymbol{x}}_j^T] \tag{104}$$

When $i \neq j$, $\bar{\boldsymbol{x}}_i$ and $\bar{\boldsymbol{x}}_j$ are independent, so

$$\mathbb{E}[\bar{\boldsymbol{x}}_i\bar{\boldsymbol{x}}_i^T\boldsymbol{A}\bar{\boldsymbol{x}}_j\bar{\boldsymbol{x}}_j^T] = \mathbb{E}[\bar{\boldsymbol{x}}_i\bar{\boldsymbol{x}}_i^T]\boldsymbol{A}\mathbb{E}[\bar{\boldsymbol{x}}_j\bar{\boldsymbol{x}}_j^T] = \boldsymbol{\Sigma}\boldsymbol{A}\boldsymbol{\Sigma} \tag{105}$$

When $i = j$,

$$\mathbb{E}[\bar{\boldsymbol{x}}_i\bar{\boldsymbol{x}}_i^T\boldsymbol{A}\bar{\boldsymbol{x}}_i\bar{\boldsymbol{x}}_i^T] = \mathbb{E}[\bar{\boldsymbol{x}}\bar{\boldsymbol{x}}^T\boldsymbol{A}\bar{\boldsymbol{x}}\bar{\boldsymbol{x}}^T] \tag{106}$$

Let $\bar{\boldsymbol{x}} = [x_1, x_2, \dots, x_d]^T$. Then, from Isserlis' theorem (Isserlis, 1918), we have

$$\mathbb{E}[x_i x_j x_k x_l] = \boldsymbol{\Sigma}_{ij}\boldsymbol{\Sigma}_{kl} + \boldsymbol{\Sigma}_{ik}\boldsymbol{\Sigma}_{jl} + \boldsymbol{\Sigma}_{il}\boldsymbol{\Sigma}_{jk} \tag{107}$$

Let $\boldsymbol{A} = [a_{ij}]$. Then, $\bar{\boldsymbol{x}}^T\boldsymbol{A}\bar{\boldsymbol{x}} = \sum_{i,j}a_{ij}x_ix_j$. Thus, we reach

$$\bar{\boldsymbol{x}}\bar{\boldsymbol{x}}^T\boldsymbol{A}\bar{\boldsymbol{x}}\bar{\boldsymbol{x}}^T = \bar{\boldsymbol{x}}\bar{\boldsymbol{x}}^T\sum_{i,j}a_{ij}x_ix_j, \tag{108}$$

$$\mathbb{E}[\bar{\boldsymbol{x}}_i\bar{\boldsymbol{x}}_i^T\boldsymbol{A}\bar{\boldsymbol{x}}_i\bar{\boldsymbol{x}}_i^T] = \mathrm{Tr}(\boldsymbol{A}\boldsymbol{\Sigma})\boldsymbol{\Sigma} + \boldsymbol{\Sigma}\boldsymbol{A}\boldsymbol{\Sigma} + \boldsymbol{\Sigma}\boldsymbol{A}^T\boldsymbol{\Sigma}. \tag{109}$$

There are $l^2$ terms in the double sum. $l$ terms are of the form $\mathbb{E}[\bar{\boldsymbol{x}}_i\bar{\boldsymbol{x}}_i^T\boldsymbol{A}\bar{\boldsymbol{x}}_i\bar{\boldsymbol{x}}_i^T]$ and $l^2 - l$ terms are of the form $\boldsymbol{\Sigma}\boldsymbol{A}\boldsymbol{\Sigma}$. Therefore, we can write

$$\mathbb{E}[\boldsymbol{S}_x\boldsymbol{A}\boldsymbol{S}_x] = \frac{1}{l^2}[l(\mathrm{Tr}(\boldsymbol{A}\boldsymbol{\Sigma})\boldsymbol{\Sigma} + \boldsymbol{\Sigma}\boldsymbol{A}\boldsymbol{\Sigma} + \boldsymbol{\Sigma}\boldsymbol{A}^T\boldsymbol{\Sigma}) + l(l-1)\boldsymbol{\Sigma}\boldsymbol{A}\boldsymbol{\Sigma}], \tag{110}$$

$$= \frac{1}{l}(\mathrm{Tr}(\boldsymbol{A}\boldsymbol{\Sigma})\boldsymbol{\Sigma} + \boldsymbol{\Sigma}\boldsymbol{A}\boldsymbol{\Sigma} + \boldsymbol{\Sigma}\boldsymbol{A}^T\boldsymbol{\Sigma}) + \frac{l-1}{l}\boldsymbol{\Sigma}\boldsymbol{A}\boldsymbol{\Sigma}, \tag{111}$$

$$= \boldsymbol{\Sigma}\boldsymbol{A}\boldsymbol{\Sigma} + \frac{1}{l}\boldsymbol{\Sigma}\boldsymbol{A}^T\boldsymbol{\Sigma} + \frac{1}{l}\mathrm{Tr}(\boldsymbol{A}\boldsymbol{\Sigma})\boldsymbol{\Sigma}, \tag{112}$$

which completes the proof. $\square$

## I    ANALYSIS OF OPTIMAL TEMPERATURE FOR ICL UNDER DISTRIBUTION SHIFT

Here, we find the optimal temperature minimizing the generalization error. First, recall that we have the following generalization error.

$$\mathcal{G}(\boldsymbol{V}, \boldsymbol{M}) = \frac{1}{\tau^2}\mathrm{Tr}\left(\boldsymbol{A}\boldsymbol{M}_{11}^T\boldsymbol{F}_1\boldsymbol{M}_{11}\right) - \frac{1}{\tau}\mathrm{Tr}\left(\boldsymbol{A}\left(\boldsymbol{F}_2\boldsymbol{M}_{11} + \boldsymbol{M}_{11}^T\boldsymbol{F}_2^T\right)\right) + \mathrm{Tr}\left(\boldsymbol{A}\boldsymbol{B}\right) + \sigma^2, \tag{113}$$

as specified in Theorem 4.6. So, we can express the generalization error as,

$$\mathcal{G}(\tau; \boldsymbol{V}, \boldsymbol{M}) = \frac{a}{\tau^2} - \frac{b}{\tau} + c, \tag{114}$$

where $a := \mathrm{Tr}\left(\boldsymbol{A}\boldsymbol{M}_{11}^T\boldsymbol{F}_1\boldsymbol{M}_{11}\right)$, $b := \mathrm{Tr}\left(\boldsymbol{A}\left(\boldsymbol{F}_2\boldsymbol{M}_{11} + \boldsymbol{M}_{11}^T\boldsymbol{F}_2^T\right)\right)$, and $c = \mathrm{Tr}\left(\boldsymbol{A}\boldsymbol{B}\right) + \sigma^2$. Therefore, we have the following optimization problem

$$\tau_{optimal} := \arg\min_{\tau} \mathcal{G}(\tau; \boldsymbol{V}, \boldsymbol{M}), \tag{115}$$

$$= \arg\min_{\tau} \left\{ \frac{a}{\tau^2} - \frac{b}{\tau} + c \right\}. \tag{116}$$

To find the optimal value of $\tau$ that minimizes the given function, we can take the derivative of the expression with respect to $\tau$ and set it to zero. From now on, we consider generalization error as a function of $\tau$, written as $\mathcal{G}(\tau)$.

Next, find the derivative of $\mathcal{G}(\tau)$ with respect to $\tau$ as

$$\mathcal{G}'(\tau) = -2a\tau^{-3} + b\tau^{-2}. \tag{117}$$

To find the critical points, set $\mathcal{G}'(\tau) = 0$ as follows

$$\mathcal{G}'(\tau) = -2a\tau^{-3} + b\tau^{-2} = 0, \tag{118}$$

Solving this equation for $\tau$, we reach the following critical point

$$\tau = \frac{2a}{b}. \tag{119}$$

Now, we need to check if this is a minimum by taking the second derivative, which is

$$\mathcal{G}''(\tau) = 6a\tau^{-4} - 2b\tau^{-3}. \tag{120}$$

Evaluate $\mathcal{G}''(\tau)$ at $\tau = \frac{2a}{b}$ as follows

$$\mathcal{G}''\left(\frac{2a}{b}\right) = 6a\left(\frac{2a}{b}\right)^{-4} - 2b\left(\frac{2a}{b}\right)^{-3} = 6a\left(\frac{b^4}{16a^4}\right) - 2b\left(\frac{b^3}{8a^3}\right) = \frac{b^4}{8a^3}. \tag{121}$$

Since $a, b > 0$, we reach $\mathcal{G}''\left(\frac{2a}{b}\right) = \frac{b^4}{8a^3} > 0$, which means the function has a minimum at $\tau = \frac{2a}{b}$. Therefore, $\tau_{optimal} = \frac{2a}{b}$ is the solution minimizing the generalization error $\mathcal{G}(\tau)$. Writing $a, b$ back into the optimal solution, we get

$$\tau_{optimal} = \frac{2\mathrm{Tr}\left(\boldsymbol{A}\boldsymbol{M}_{11}^T\boldsymbol{F}_1\boldsymbol{M}_{11}\right)}{\mathrm{Tr}\left(\boldsymbol{A}\left(\boldsymbol{F}_2\boldsymbol{M}_{11} + \boldsymbol{M}_{11}^T\boldsymbol{F}_2^T\right)\right)}, \tag{122}$$

which concludes our derivation of the optimal temperature $\tau_{optimal}$.

## J  AN INSIGHT DRIVEN FROM OPTIMAL TEMPERATURE FOR OTHER SETTINGS

In this section, we extract a mathematical heuristic from the optimal temperature in Theorem 4.7 that can be applied to ICL settings beyond our existing setting involving linearized attention and regression tasks. Specifically, we consider Transformers employing standard softmax attention. Recall that the attention temperature scales the pre-softmax scores (i.e., $(\boldsymbol{K}\boldsymbol{Z})^\top(\boldsymbol{Q}\boldsymbol{Z})$ in (1)), thereby controlling the variance of the final scores. Since the optimal temperature depends on the distribution of these scores, it can be naturally characterized by the moments of that distribution. Our central intuition is that the optimal temperature identified in Theorem 4.7 relates directly to the first two moments of the pre-softmax scores. Although this optimal temperature was derived for *linearized softmax* attention, the insight remains relevant for softmax attention because the two mechanisms behave similarly in the regime considered (see Appendix D).

We now illustrate how the optimal temperature in Theorem 4.7 can be related to the first two moments of the pre-softmax scores. For simplicity, we consider the case $\boldsymbol{\mu}_x = \boldsymbol{\mu}_w = \boldsymbol{m}_{21} = \boldsymbol{0}$ and $\boldsymbol{\Sigma}_w = \boldsymbol{I}$, under which the optimal temperature reduces to

$$\tau_{\text{optimal}} = \frac{v_{22}\,\text{Tr}\big(\boldsymbol{\Sigma}_x \boldsymbol{M}_{11} \boldsymbol{\Sigma}_x \boldsymbol{M}_{11}^\top \boldsymbol{\Sigma}_x\big)}{\frac{1}{2}\,\text{Tr}\big(\boldsymbol{\Sigma}_x \big(\boldsymbol{\Sigma}_x \boldsymbol{M}_{11} + \boldsymbol{M}_{11}^\top \boldsymbol{\Sigma}_x^\top\big)\big)}. \tag{123}$$

We next show how this expression connects to the first two moments of $(\boldsymbol{KZ})^\top(\boldsymbol{QZ})$. Let $\boldsymbol{z}_i$ denote the $i$-th column of $\boldsymbol{Z}$ from (3) and recall $\boldsymbol{K}^\top \boldsymbol{Q} = \boldsymbol{M}$. We therefore compute $\mathbb{E}[\boldsymbol{z}_i^\top \boldsymbol{M} \boldsymbol{z}_j]$ and $\mathbb{E}[(\boldsymbol{z}_i^\top \boldsymbol{M} \boldsymbol{z}_j)^2]$ for $i, j \in \{1, \ldots, l\}$. Starting with the first moment for $i = j$:

$$\mathbb{E}[\boldsymbol{z}_i^\top \boldsymbol{M} \boldsymbol{z}_i] = \text{Tr}\big(\mathbb{E}[\boldsymbol{z}_i \boldsymbol{z}_i^\top]\boldsymbol{M}\big) = \text{Tr}\big(\boldsymbol{\Sigma}_x \boldsymbol{M}_{11}\big), \tag{124}$$

where the block structure (and zero entries) of $\boldsymbol{M}$ is used in the last step. For $i \neq j$,

$$\mathbb{E}[\boldsymbol{z}_i^\top \boldsymbol{M} \boldsymbol{z}_j] = \text{Tr}\big(\boldsymbol{M}\mathbb{E}[\boldsymbol{z}_i \boldsymbol{z}_j^\top]\big) = 0, \tag{125}$$

by independence of $\boldsymbol{z}_i$ and $\boldsymbol{z}_j$. For the second moment with $i \neq j$:

$$\mathbb{E}\big[(\boldsymbol{z}_i^\top \boldsymbol{M} \boldsymbol{z}_j)^2\big] = \mathbb{E}\big[\boldsymbol{z}_i^\top \boldsymbol{M} \boldsymbol{z}_j \boldsymbol{z}_j^\top \boldsymbol{M}^\top \boldsymbol{z}_i\big] \tag{126}$$

$$= \mathbb{E}\big[\boldsymbol{x}_i^\top \boldsymbol{M}_{11} \boldsymbol{x}_j \boldsymbol{x}_j^\top \boldsymbol{M}_{11}^\top \boldsymbol{x}_i\big] \tag{127}$$

$$= \mathbb{E}_{\boldsymbol{x}_i}\big[\boldsymbol{x}_i^\top \boldsymbol{M}_{11} \mathbb{E}_{\boldsymbol{x}_j}[\boldsymbol{x}_j \boldsymbol{x}_j^\top] \boldsymbol{M}_{11}^\top \boldsymbol{x}_i\big] \tag{128}$$

$$= \mathbb{E}_{\boldsymbol{x}_i}\big[\boldsymbol{x}_i^\top \boldsymbol{M}_{11} \boldsymbol{\Sigma}_x \boldsymbol{M}_{11}^\top \boldsymbol{x}_i\big] \tag{129}$$

$$= \text{Tr}\big(\boldsymbol{M}_{11} \boldsymbol{\Sigma}_x \boldsymbol{M}_{11}^\top \mathbb{E}_{\boldsymbol{x}_i}[\boldsymbol{x}_i \boldsymbol{x}_i^\top]\big) \tag{130}$$

$$= \text{Tr}\big(\boldsymbol{M}_{11} \boldsymbol{\Sigma}_x \boldsymbol{M}_{11}^\top \boldsymbol{\Sigma}_x\big), \tag{131}$$

where we again exploit the block structure of $\boldsymbol{M}$ and apply straightforward manipulations.

We observe a parallel between the numerator of (123) and the computed second moment (for $i \neq j$), and between the denominator and the first moment (for $i = j$). This motivates the heuristic that the optimal temperature should be roughly proportional to the ratio of the second moment (for $i \neq j$) to the first moment (for $i = j$). Accordingly, in our LLM experiments (Figure 3), we select the temperature proportional to this ratio while taking care to avoid numerical issues.

Finally, we note an important caveat: in order to obtain an insight of practical relevance, we intentionally relaxed the rigor applied in our main theoretical results. Consequently, the heuristic derived here—and the accompanying empirical findings—should be viewed as preliminary, intended to inspire future work on principled selection of attention temperature in practice.

## K  Experimental details and GPT-2 experiments

This section describes our experimental setups for GPT-2 and large language models (LLMs), including the motivation for our distribution-shift scenarios.

### K.1  GPT-2: Transformer with MLP layers

Building on the linearized-attention experiments, we investigate whether the optimal temperature also benefits more complex Transformer models on linear regression tasks. We evaluate GPT-2 (Radford et al., 2019) under a shift in input covariance (Figure 6). Consistent with prior work (Garg et al., 2022; Zhang et al., 2024), such shifts substantially degrade performance and can even induce non-monotonic generalization error with respect to context length $l$. Remarkably, applying the optimal temperature mitigates this nonmonotonicity and improves in-context generalization.

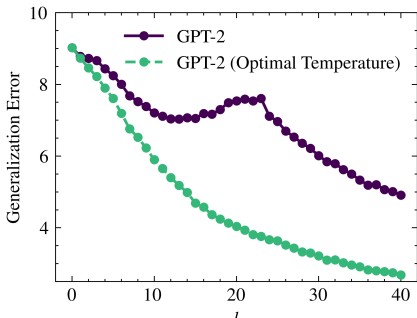

Figure 6: GPT-2 (Radford et al., 2019) under an input-covariance shift. GPT-2 exemplifies the Transformer architecture (Vaswani et al., 2017), combining multi-layer perceptrons with multi-head softmax self-attention. The model here is pretrained by Garg et al. (2022) on the linear regression tasks defined in (2). We consider a shift from $\mathbf{\Sigma}_x^{\text{train}} = \boldsymbol{I}$ to $\mathbf{\Sigma}_x^{\text{test}} = 3\boldsymbol{I}$. The attention temperature at each layer is scaled as $\tau\sqrt{d_k}$ (where $d_k$ is the key dimension) to ensure dimension-independent $\tau$ values.

### K.2 Details of the GPT-2 experiments in Figure 6

We use the standard GPT-2 architecture (Radford et al., 2019) as implemented in HuggingFace (Wolf et al., 2020), leveraging the pretrained model of Garg et al. (2022). Training data match ours, while their training procedure differs slightly: the loss is auto-regressive, i.e., the average over the entire context sequence of length $l = 40$. We adopt the same embedding method as in Garg et al. (2022). The input dimension is $d = 20$, with 12 layers and 8 heads. All GPT-2 experiments run on an NVIDIA Tesla V100 GPU and complete in approximately 10 minutes.

### K.3 Details of the LLM experiments in Figure 3

For our large language model experiments, we employ LLaMA2-7B (Touvron et al., 2023) and the SCIQ dataset (Welbl et al., 2017), which contains science questions with supporting information. We generate ICL problems following Gao et al. (2024), selecting in-context demonstrations using the TopK retrieval technique (Liu et al., 2022) to ensure relevance. An example ICL sample from SCIQ appears in Table 1. To simulate distribution shift, we follow Gao et al. (2024) and introduce noisy labels—incorrect but semantically related—to the in-context demonstrations (Appendix K.4). Table 2 gives an example. The noisy ratio denotes the fraction of demonstrations with noisy labels (e.g., 0.6 means 60% noisy). We modify and use the codebase of Gao et al. (2024), built on HuggingFace (Wolf et al., 2020) and OpenICL (Wu et al., 2023). All LLM experiments run on an NVIDIA A40 GPU; a single Monte Carlo run per plot in Figure 3 takes a few hours.

### K.4 Why in-context demonstrations with noisy labels as an example of distribution shift?

The link between noisy labels in demonstrations and distribution shift may not be immediately obvious. Quantifying pretraining–test shifts for pretrained LLMs is inherently difficult because their pretraining data are complex mixtures of sources (Touvron et al., 2023). However, we hypothesize—following Gao et al. (2024)—that high perplexity can serve as an empirical indicator of distribution shift. Inputs aligned with the training distribution tend to yield low perplexity (high-confidence generation), whereas contradictory or out-of-distribution inputs induce high perplexity. Since noisy demonstrations are expected to contradict training-set patterns, they yield high perplexity and thereby act as a proxy for distribution shift. Consequently, introducing noisy labels into in-context demonstrations constitutes a principled way to test the robustness of in-context learning under distribution shift.

| **In-context demonstration 1** | |
|---|---|
| **Support:** | Cells are organized into tissues, tissues are organized into organs. |
| **Question:** | What is considered the smallest unit of the organ? |
| **Answer:** | Cells |

| **In-context demonstration 2** | |
|---|---|
| **Support:** | . . . four basic types of tissue: connective, muscle, nervous, and epithelial. |
| **Question:** | The four basic types of tissue are epithelial, muscle, connective, and what? |
| **Answer:** | nervous |

⋮

| **Test example** | |
|---|---|
| **Support:** | All forms of life are built of at least one cell. A cell is the basic unit of life. |
| **Question:** | What are the smallest structural and functional units of all living organisms? |
| **Output:** | ??? |

Table 1: A sample illustration of in-context learning on the SCIQ dataset.

| Setting | In-context demonstration |
|---|---|
| True Label | **Support:** Cells are organized into tissues, tissues are organized into organs. 
 **Question:** What is considered the smallest unit of the organ? 
 **Label:** Cells |
| Noisy Label | **Support:** Cells are organized into tissues, tissues are organized into organs. 
 **Question:** What is considered the smallest unit of the organ? 
 **Label:** tissues |

Table 2: An example of a true label vs. a relevant but noisy label. A relevant label is related to the question but is not necessarily true. Therefore, relevant labels can be considered noisy labels.

## L    REST OF THE RELATED WORK

**ICL by Transformers —**    The ICL capability of Transformers was first brought to prominence by Brown et al. (2020), leading to a surge of empirical and theoretical investigations. Several works have demonstrated that ICL performance improves with model scale Wei et al. (2022); Olsson et al. (2022); Schaeffer et al. (2023), underscoring its importance in modern AI systems. To better understand this phenomenon, synthetic tasks such as linear regression have served as controlled testbeds for analyzing ICL in Transformers (Garg et al., 2022; Zhang et al., 2024; Raventós et al., 2023). A prevailing hypothesis in recent theoretical work is that Transformers implicitly learn algorithms during pretraining, which they subsequently execute during inference (Bai et al., 2023; Li et al., 2023; Akyürek et al., 2023; Ahn et al., 2023; Von Oswald et al., 2023; Mahankali et al., 2024; Fu et al., 2024; Zhang et al., 2024; Li et al., 2024; Park et al., 2024). There remains ongoing debate over the precise nature of these learned procedures. However, our work focuses on a fundamentally different question, which is how attention temperature affects the ICL performance of pretrained Transformers under distribution shifts.

## M    INTERPRETING THE CLOSED-FORM OPTIMAL TEMPERATURE: ANALYTICAL REDUCTIONS AND EMPIRICAL VALIDATION

This section provides an expanded and more interpretable discussion of the optimal attention temperature derived in Theorem 4.7. Our goal is to illustrate how the closed-form expression behaves

under concrete distribution shifts and to clarify its relationship to the moment-ratio heuristic introduced in Appendix J.

## M.1    ANALYTICAL SPECIALIZATION OF THEOREM 4.7

To obtain a simplified expression, we consider a simple but representative family of shifts as follows. The training distributions are

$$\text{(input) } \boldsymbol{x}_i \sim \mathcal{N}(\mathbf{0}, \boldsymbol{I}), \qquad \text{(task) } \boldsymbol{w} \sim \mathcal{N}(\mathbf{0}, \boldsymbol{I}), \qquad \text{(noise) } \epsilon_i \sim \mathcal{N}(0, 0.1^2).$$

We then introduce three independent shift parameters for the test distribution:

$$\text{(input) } \boldsymbol{x}_i \sim \mathcal{N}(\mathbf{0}, a\boldsymbol{I}), \qquad \text{(task) } \boldsymbol{w} \sim \mathcal{N}(\mathbf{0}, b\boldsymbol{I}), \qquad \text{(noise) } \epsilon_i \sim \mathcal{N}(0, \sigma^2),$$

where $a > 0$ controls the input variance shift, $b > 0$ controls the task-parameter variance shift, and $\sigma > 0$ controls the noise-variance shift. This setting preserves isotropy, which makes it possible to derive a clean closed-form expression while still connecting directly to realistic distribution shifts.

Substituting these shifted distributions into the optimal-temperature expression in (15) yields

$$\tau_{\text{optimal}} = \frac{2\text{Tr}\left(a\boldsymbol{I}\boldsymbol{M}_{11}^T \left(ab\boldsymbol{I} + \frac{1}{l}(\sigma^2 + abd)\boldsymbol{I}\right) a\boldsymbol{I}\boldsymbol{M}_{11}\right)}{\text{Tr}\left(a\boldsymbol{I}\left(ab\boldsymbol{I}\left(\boldsymbol{M}_{11} + \boldsymbol{M}_{11}^T\right)\right)\right)}, \tag{132}$$

$$= \left(a + \frac{1}{l}\left(\frac{\sigma^2}{b} + ad\right)\right)\frac{\text{Tr}\left(\boldsymbol{M}_{11}^T\boldsymbol{M}_{11}\right)}{\text{Tr}\left(\boldsymbol{M}_{11}\right)}. \tag{133}$$

This concrete formula makes several effects fully explicit:

- *Input shift* — Increasing input variance $a$ directly scales $\tau_{\text{optimal}}$ upward. This aligns with our earlier results (Figure 1) and the heuristic derived in Appendix J, which suggests that a greater variance of pre-softmax scores requires a higher temperature to maintain robustness to input shifts.
- *Noise shift* — Increasing noise variance $\sigma^2$ also increases $\tau_{\text{optimal}}$, but only through the $\frac{1}{l}$ term, reflecting the diminishing effect of noise when more in-context examples are available.
- *Task shift* — Increasing task variance $b$ reduces the effect of noise (via $\sigma^2/b$), slightly lowering the optimal temperature.
- *Context length* — As $l \to \infty$, the $\frac{1}{l}$ term vanishes, giving a simplified asymptotic rule:

$$\tau_{\text{optimal}} \to a \cdot \frac{\text{Tr}(\boldsymbol{M}_{11}^\top\boldsymbol{M}_{11})}{\text{Tr}(\boldsymbol{M}_{11})}.$$

Note that under the considered training distribution, we have $\text{Tr}(\boldsymbol{M}_{11}^\top\boldsymbol{M}_{11})/\text{Tr}(\boldsymbol{M}_{11}) \approx 1$, which implies that $\tau_{\text{optimal}} \to a$ as $l \to \infty$. Thus, even in this simplified scenario, the optimal temperature explicitly tracks the magnitude and nature of the distribution shifts.

## M.2    CONNECTION TO THE MOMENT-RATIO HEURISTIC

Recall that Appendix J proposed a practical heuristic based on the ratio of the second and first moments of pre-softmax attention scores. The expression in (133) provides further theoretical justification for that heuristic.

Indeed, for some $i \neq j$,

$$\tau_{optimal} = a\frac{\text{Tr}\left(\boldsymbol{M}_{11}\boldsymbol{M}_{11}^T\right)}{\text{Tr}\left(\boldsymbol{M}_{11}\right)} + \frac{1}{l}\left(\frac{\sigma^2}{b} + ad\right)\frac{\text{Tr}\left(\boldsymbol{M}_{11}\boldsymbol{M}_{11}^T\right)}{\text{Tr}\left(\boldsymbol{M}_{11}\right)}, \tag{134}$$

$$= \frac{\mathbb{E}\left[(\boldsymbol{z}_i^\top\boldsymbol{M}\boldsymbol{z}_j)^2\right]}{\mathbb{E}\left[\boldsymbol{z}_i^\top\boldsymbol{M}\boldsymbol{z}_i\right]} + \frac{1}{l}\left(\frac{\sigma^2}{b} + ad\right)\frac{\text{Tr}\left(\boldsymbol{M}_{11}\boldsymbol{M}_{11}^T\right)}{\text{Tr}\left(\boldsymbol{M}_{11}\right)}, \tag{135}$$

where we used the moments calculated in Appendix J to reach the final line. Here, since $\text{Tr}(\boldsymbol{M}_{11}^\top\boldsymbol{M}_{11})/\text{Tr}(\boldsymbol{M}_{11}) \approx 1$ for the considered training distribution, this gives the approximation:

$$\tau_{\text{optimal}} \approx \underbrace{\frac{\mathbb{E}\left[(\boldsymbol{z}_i^\top\boldsymbol{M}\boldsymbol{z}_j)^2\right]}{\mathbb{E}\left[\boldsymbol{z}_i^\top\boldsymbol{M}\boldsymbol{z}_i\right]}}_{\text{moment-ratio}} + \underbrace{\frac{1}{l}\left(\frac{\sigma^2}{b} + ad\right)}_{\text{correction for small } l}. \tag{136}$$

This demonstrates that the moment-ratio heuristic is not an ad-hoc rule, but a theoretically grounded approximation of the exact closed-form optimal temperature.

### M.3   NUMERICAL ILLUSTRATION OF THE OPTIMAL TEMPERATURE AND ITS GENERALIZATION BEHAVIOR

To complement the analytical reductions above, we now present numerical experiments illustrating how the optimal attention temperature varies under different types of distribution shift. These simulations closely follow the structure predicted by the closed-form expression in Theorem 4.7 and its simplified forms in (133) and (136).

Figure 7 shows the optimal temperature as a function of (i) input covariance shift, (ii) task covariance shift, and (iii) noise-variance shift. In each subplot, a single shift parameter ($a$, $b$, or $\sigma$) is varied while the others remain fixed. The closed-form optimal temperatures (15) align closely with the moment-ratio heuristic with correction (136). As anticipated:

- higher input variance $a$ or noise level $\sigma$ increases the optimal temperature, while
- task variance $b$ does not significantly change the optimal temperature.

Figure 8 presents the corresponding generalization errors. These results demonstrate that the closed-form characterization accurately captures the key dependencies of the optimal temperature under a range of distribution shifts.

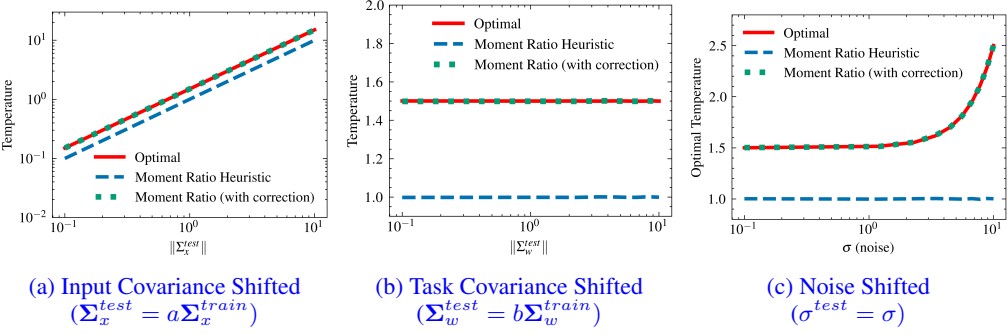

(a) Input Covariance Shifted ($\boldsymbol{\Sigma}_x^{test} = a\boldsymbol{\Sigma}_x^{train}$)

(b) Task Covariance Shifted ($\boldsymbol{\Sigma}_w^{test} = b\boldsymbol{\Sigma}_w^{train}$)

(c) Noise Shifted ($\sigma^{test} = \sigma$)

Figure 7: Optimal temperature under different types of distribution shift. The moment-ratio heuristic (Appendix J) is derived from the closed-form optimal temperature, with its corrected form given in (136). During training, we use $m = 5000$ tasks, noise level $\sigma^{train} = 0.1$, means $\boldsymbol{\mu}_x^{train} = \boldsymbol{\mu}_w^{train} = \mathbf{0}$, and covariances $\boldsymbol{\Sigma}_x^{train} = \boldsymbol{\Sigma}_w^{train} = \boldsymbol{I}$. At test time, we set $\boldsymbol{\mu}_x^{test} = \boldsymbol{\mu}_w^{test} = \mathbf{0}$, $\boldsymbol{\Sigma}_x^{test} = aI$, $\boldsymbol{\Sigma}_w^{test} = bI$, and $\sigma^{test} = \sigma$. In each subplot, exactly one of $a$, $b$, or $\sigma$ is varied, while the other two remain fixed at their training-distribution values to isolate the effect of a single shift dimension. The dimension and context length are set to $d = 50$ and $l = 2d$.

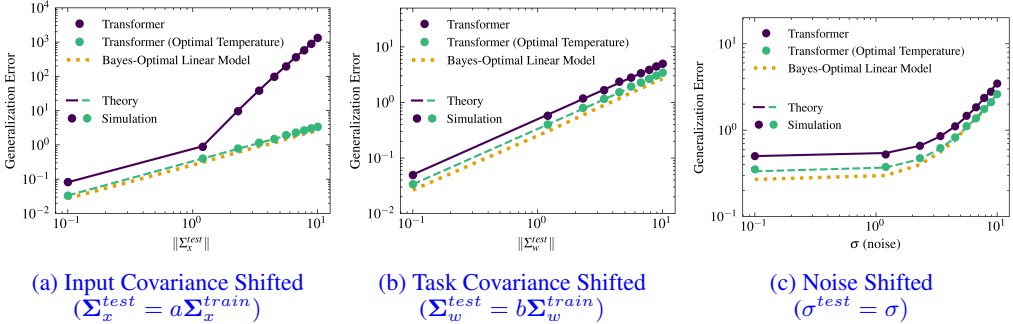

(a) Input Covariance Shifted ($\boldsymbol{\Sigma}_x^{test} = a\boldsymbol{\Sigma}_x^{train}$)

(b) Task Covariance Shifted ($\boldsymbol{\Sigma}_w^{test} = b\boldsymbol{\Sigma}_w^{train}$)

(c) Noise Shifted ($\sigma^{test} = \sigma$)

Figure 8: Generalization errors corresponding to Figure 7.

