# OpenReview forum: "Optimal Attention Temperature Enhances In-Context Learning under Distribution Shift"
_ICLR.cc/2026/Conference — Submitted to ICLR 2026_

### Official Review · Reviewer_2yfj · 2025-10-28

**Soundness:** 3
**Presentation:** 3
**Contribution:** 2
**Rating:** 4
**Confidence:** 4

**Summary:**

This paper theoretically investigates how adjusting the attention temperature (τ) within the softmax attention mechanism of Transformers affects in-context learning (ICL), especially under distribution shift between pretraining and test data. Unlike prior works that fix τ, it demonstrates that an optimal τ exists that minimizes generalization error — thereby improving ICL robustness.

**Strengths:**

- Theoretical derivation is clear and matches empirical behavior.
- Simple insight (temperature tuning) leads to measurable robustness gains.
- Bridges the gap between theory and practical tuning of LLMs.

**Weaknesses:**

1. Temperature tuning seems to be a well-known method, and the contribution of this work seems to be limited to the theoretical analysis on a simplified linear regression model.

2. While the authors derive a closed-form of the optimal temperature of the linear regression case, the insight into how to tune the parameter and the quantitative analysis of the effect of the temperature are still missing.

3. Section 4.2 is confusing. The discussion of different cases seems to be made without sufficient evidence, and I do not think Lemma 4.1 and Corollary 4.4 can support the discussion on distribution shift cases.

**Questions:**

See weakness.

---

> ### Author Response · Authors · 2025-11-19
>
> We thank the reviewer for evaluating our paper. We appreciate your recognition of several key strengths: (i) the clarity of the theoretical derivation and its agreement with empirical behavior, (ii) the robustness gains enabled by temperature tuning, and (iii) the connection our work establishes between theory and practical LLM tuning. Before addressing the individual comments, we would like to reiterate that the paper’s primary contribution is theoretical: it provides the first analytical characterization of how attention temperature influences ICL generalization error under distribution shifts and derives the corresponding optimal temperature in closed form. We hope this perspective helps frame the novelty of the results as we address your points below.
>
> > Temperature tuning seems to be a well-known method, and the contribution of this work seems to be limited to the theoretical analysis on a simplified linear regression model.
>
> Thank you for raising this point. **However, the concern that "temperature tuning is well-known" does not apply to *attention temperature* for the case of in-context learning (ICL). To the best of our knowledge, the role of attention temperature in Transformers has *not* been studied—neither empirically nor theoretically—for ICL or for settings involving distribution shift. Our work fills precisely this gap.**
>
> Moreover, our results provide the first theoretical characterization of how attention temperature affects ICL generalization error, and the first analytic expression for an optimal attention temperature under distribution shift. These contributions go beyond the simplified model itself: they reveal a principled mechanism by which test-time temperature interacts with score statistics and distributional changes.
>
> We also emphasize that attention temperature is **_fundamentally distinct from the well-known sampling temperature_** used to adjust the output distribution of generative models. While sampling temperature has been widely discussed, *tuning the attention temperature is not a standard technique*, and its effect on ICL behavior has not been previously studied.
>
> > While the authors derive a closed-form of the optimal temperature of the linear regression case, the insight into how to tune the parameter and the quantitative analysis of the effect of the temperature are still missing.
>
> Thank you for the comment. **We would like to clarify that our paper *does* provide concrete insight into how to choose the optimal attention temperature in Sections 4.4-5.2 and Appendix J**. We explain that the optimal temperature increases when either the input variance or the noise variance increases in our setting. Furthermore, **during the rebuttal, we expanded our discussion regarding the interpretation of the closed-form solution for the optimal temperature *by delivering new analytical and experimental results* in Appendix M, which is located at the end of the paper.** This new appendix includes additional analytical explanations and experimental results regarding the optimal attention temperature.
>
> More generally, based on the found closed-form of the optimal attention temperature, we derive **a practical insight (discussed in Section 5.2 and Appendix J in the initial submission), which is that the optimal $\tau$ is proportional to the *variance-to-mean ratio of the pre-softmax scores*.** This yields a practical, data-dependent rule for tuning the temperature. We further validate this insight in our LLM experiments (Figure 3), where the same variance-to-mean–based heuristic leads to improved robustness under distribution shift.

---

> > ### Author Response · Authors · 2025-11-19
> >
> > > Section 4.2 is confusing. The discussion of different cases seems to be made without sufficient evidence, and I do not think Lemma 4.1 and Corollary 4.4 can support the discussion on distribution shift cases.
> >
> > Thank you for the feedback. We are happy to clarify the role of Section 4.2. Lemma 4.1 provides the pretrained parameters mimicking the Bayes-optimal under the *training* distribution. In Section 4.2, **we examine how these pretrained parameters would differ if the model had instead been trained on the *shifted test-time distribution*.** This comparison—formalized using Lemma 4.1 and Corollary 4.4—shows precisely when the distribution shift renders the pretrained parameters significantly suboptimal.
> >
> > The key idea is that such deviations between the pretrained parameters and the optimal parameters for the test distribution directly predict reduced generalization performance. Hence, **this approach allows us to identify which kinds of distribution shifts (e.g., changes in $\Sigma_x$, $\Sigma_w$, or $\sigma^2$) matter for in-context learning.** The cases in Section 4.2 follow systematically from this analysis, and we will revise the text to make these connections more explicit.
> >
> > ---
> >
> > Overall, our paper presents foundational results on optimal attention temperature for ICL under distribution shifts, offering precise analytical characterizations of both ICL performance and the corresponding optimal temperature. **In the rebuttal period, our new analytical and experimental additions in Appendix M sharpened the interpretability of our theoretical results (Theorems 4.6–4.7), and our responses clearly articulated the work’s theoretical contribution and novelty.** We believe we have fully addressed the concerns you raised, and we respectfully ask you to reconsider your evaluation in light of these clarifications.

---

> ### Author Response · Authors · 2025-11-26
>
> Dear Reviewer 2yfj, we would like to kindly follow up to ask whether our expanded explanation of the practical temperature-selection heuristic (based on the variance-to-mean ratio), the additional analysis in Appendix M, and the clarifications to Section 4.2 address your concerns about insight into optimal temperature choice and the support for the distribution-shift cases. If there are remaining issues, we would highly appreciate further questions or your reconsideration of the evaluation.

---

### Official Review · Reviewer_h3fo · 2025-10-30

**Soundness:** 3
**Presentation:** 2
**Contribution:** 2
**Rating:** 4
**Confidence:** 4

**Summary:**

This paper theoretically characterizes the optimal attention temperature of Transformers with linear attention in ICL tasks. The authors analyze the generalization error when different kinds of distribution shift exist during testing and show that the optimal temperature can mitigate the impact of distributional shifts. Experiments are conducted to support the theory.

**Strengths:**

1. The proposed research about the optimal attention temperature is interesting and well-motivated.

2. The theoretical results are solid.

**Weaknesses:**

1. Theorems 4.6 and 4.7 do not theoretically indicate how the generalization error and the optimal temperature are related to the distribution shifts. There is no discussion of how Eqn. (15) will change if some parameters of the distribution change.

2. I think the basic setting of linear attention by the simplification of Eqn. (7) is well studied in existing works. Only adding the learnable temperature is not theoretically challenging enough. This makes the theoretical contribution less significant. It is better to summarize the theoretical novelty of this work.

**Questions:**

1. If my understanding is correct, you set $\tau=1$ for pretraining, and a changable $\tau$ for testing. If so, why do you set $\tau=1$ for pretraining? Why not set $\tau$ as the temperature during pretraining and $\tau'\neq \tau$ as the temperature during testing?

2. In line 411, why does the optimal temperature $\tau\_{optimal}=c$ fully counteract the shift?

3. In line 453, you say, "noise effects diminish as the context length increases, consistent with our theoretical predictions." Which theorem do you refer to? Do you mean the $\sigma/l$ terms in Eqn. 11? If so, please specify which equation the experiment verifies in this statement.

4. What does Figure 3(a) imply? What is the trade-off between added context and accumulated noise by line 471? How is it related to your theory?

---

> ### Author Response · Authors · 2025-11-19
>
> We thank the reviewer for taking the time to evaluate our paper. We appreciate your recognition that the study of optimal attention temperature is interesting and well-motivated, and we are glad that you found the theoretical results to be solid. Before addressing the specific concerns, we emphasize that our work is primarily a theoretical study, precisely characterizing the generalization error for ICL under distribution shift and deriving the optimal attention temperature in closed form. We hope this perspective helps contextualize the novelty of our results as we address your comments below.
>
>
> > Theorems 4.6 and 4.7 do not theoretically indicate how the generalization error and the optimal temperature are related to the distribution shifts. There is no discussion of how Eqn. (15) will change if some parameters of the distribution change.
>
> We appreciate the opportunity to clarify this point. **Theorems 4.6 and 4.7 directly encode how both the generalization error and the optimal temperature depend on the test-time distribution**, since the data-distribution parameters (e.g., $\Sigma_x$, $\Sigma_w$, $\sigma^2$) enter the matrices $A$, $B$, $F_1$, and $F_2$ in Eq. (12–15), while the learned model parameters (e.g., $M_{11}$) reflect the pretraining distribution. This explicit separation is what allows the analysis to capture the effect of distribution shifts. For example:
>
> – **Input covariance shift** modifies $A=\Sigma_x+\mu_x\mu_x^\top$, influencing both numerator and denominator in the closed-form optimal temperature (Eq. 15);
>
> – **Task-distribution shift** affects $B$, $\hat{B}$ and $F_2$, altering the numerator and denominator;
>
> – **Noise shift** enters through the $\sigma^2$ term in $F_1$, changing the numerator.
>
> In addition to this theoretical mapping, **we further derive a practical insight based on Theorem 4.7 in Section 5.2 and Appendix J**, where we show that the optimal temperature is approximately proportional to the variance-to-mean ratio of the pre-softmax scores. **To improve interpretability, we also added Appendix M during the rebuttal, which provides new analytical and experimental illustrations of how the optimal attention temperature changes under various distribution shifts.**
>
>
> > I think the basic setting of linear attention by the simplification of Eqn. (7) is well studied in existing works. Only adding the learnable temperature is not theoretically challenging enough. This makes the theoretical contribution less significant. It is better to summarize the theoretical novelty of this work.
>
>
> We respectfully disagree with the impression that our setting is only a minor extension of prior linear-attention analyses. While previous works (e.g., Zhang et al., 2024) study *purely linear attention*, **our analysis focuses on a softmax-like mechanism: the linearized softmax retains the key temperature-dependent effects of softmax (please see Appendix D)**, which fundamentally changes the behavior of the attention mechanism and enables studying effects that cannot be captured by linear attention. This distinction introduces **substantial additional structure and complexity** into the generalization-error analysis and the derivation of the optimal temperature.
>
> Furthermore, to analyze temperature–shift interactions rigorously, **our assumptions are intentionally broader** than those in prior linear-attention work. For instance, Assumption 3.1 accommodates general data and task distributions, and these richer settings introduce several nontrivial terms in the closed-form expressions for the generalization error.
>
> Taken together, these points yield two theoretical contributions that do not appear in prior work:
> (i) **the first closed-form characterization of ICL generalization error under distribution shift for a softmax-like attention mechanism**, and
> (ii) **the first analytic expression for an optimal attention temperature**.
> We will make these contributions more explicit in the final version.

---

> > ### Author Response · Authors · 2025-11-19
> >
> > > If my understanding is correct, you set $\tau = 1$ for pretraining, and a changable $\tau$ for testing. If so, why do you set $\tau=1$ for pretraining? Why not set $\tau$ as the temperature during pretraining and $\tau^\prime = \tau$ as the temperature during testing?
> >
> > Thank you for the question. *During pretraining, the effect of using a temperature $\tau \neq 1$ can be absorbed entirely into the learned model parameters (e.g., the matrices $M$ and $V$).* In other words, scaling the attention logits by $\tau$ during training is equivalent to a reparameterization of these weights, and therefore does not yield any additional modeling power. For this reason, it is without loss of generality to set $\tau = 1$ during pretraining.
> >
> > In contrast, at test time the model parameters are fixed, and distribution shift alters the score statistics. In this setting, the attention temperature becomes a meaningful degree of freedom, and adjusting $\tau$ provides a principled form of adaptation, as captured by our derivation of the optimal test-time temperature.
> >
> >
> > > In line 411, why does the optimal temperature $\tau_{optimal} =c $ fully counteract the shift?
> >
> > Thank you for requesting clarification. In that part of the analysis, we consider a specific distribution-shift scenario in which only the input distribution changes from $\mathcal{N}(0, I)$ to $\mathcal{N}(0, cI)$, while the task distribution remains $\mathcal{N}(0, I)$ and the noise level is fixed at $\sigma = 0$. *Substituting these shifted statistics into Eq. (15) (in Theorem 4.7) yields $\tau_{\mathrm{opt}} = c$ as $l \to \infty$.* In this special case, the shift uniformly rescales all pre-softmax scores, and the optimal temperature exactly cancels this scaling—hence fully counteracting the shift. For more information, please refer to our new results in Appendix M, including an extended discussion of the closed-form optimal temperature equation (Eq. 15).
> >
> >
> > > In line 453, you say, "noise effects diminish as the context length increases, consistent with our theoretical predictions." Which theorem do you refer to? Do you mean the $\sigma/l$ terms in Eqn. 11? If so, please specify which equation the experiment verifies in this statement.
> >
> > Thank you for pointing this out. *Yes, the statement refers to the $\sigma^2 / l$ dependence that appears in Eq. (11).* More formally, the diminishing effect of noise with increasing context length follows from Lemma 4.11 (as discussed in Section 4.2, Case III) and is also reflected in the general expressions for the ICL error in Theorems 4.6–4.7, particularly Eq. (13). We will add an explicit reference to these equations in the revised version.
> >
> > > What does Figure 3(a) imply? What is the trade-off between added context and accumulated noise by line 471? How is it related to your theory?
> >
> > Thanks for the question. *Figure 3(a) provides an initial empirical illustration of how applying the optimal attention temperature can improve ICL performance as the context length varies*, reflecting the qualitative trend suggested by our theory. *Figure 3(b) shows that higher noise levels tend to correspond to larger optimal temperatures*, again consistent with the behavior predicted by our closed-form expression (Theorem 4.7). These experimental results are intended as a first practical glimpse of the theoretical insights. Specifically, Figures 3(a) and 3(b) for the LLM experiments are analogous to Figures 2(a) and 2(b) under our controlled theoretical setting. We clarified this point in the revised paper.
> >
> > ---
> >
> > Overall, our paper establishes foundational results on the optimal attention temperature for in-context learning under distribution shifts, providing precise analytical characterizations of both ICL performance and the corresponding optimal temperature. **During the rebuttal period, our new analytical and experimental results in Appendix M improved the interpretability of our theoretical results (Theorem 4.6-4.7), while our rebuttal comments clarified the theoretical contribution/novelty of our work.** We believe we have thoroughly addressed all of your concerns, and we would greatly appreciate it if you could reconsider your evaluation in light of these clarifications.

---

> > > ### Author Response · Authors · 2025-11-26
> > >
> > > Dear Reviewer h3fo, we would like to briefly follow up to check whether our expanded discussion of Eq. (15) and the new analytical/experimental results in Appendix M clarify how the optimal temperature depends on different types of distribution shift and help address your concerns about the theoretical novelty. If not, we would be very happy to provide further details and would be grateful if you could reconsider your current evaluation in light of these additions.

---

### Official Review · Reviewer_mUBM · 2025-10-31

**Soundness:** 3
**Presentation:** 3
**Contribution:** 2
**Rating:** 4
**Confidence:** 3

**Summary:**

This paper analyzes how the attention temperature at inference time affects in-context learning (ICL) performance when there is a distribution shift between pretraining and test data (in the input, task, or noise distributions).
Using a simplified Transformer with linearized softmax attention, the authors derive a closed-form expression for the ICL generalization error as a function of the model parameters and the attention temperature.
They also provide an analytic construction of the model parameters that makes this simplified Transformer approximate the Bayes-optimal predictor for the training distribution (the “train-Bayes optimal”).
Under this setting, they show that adjusting the attention temperature at test time can provably minimize the generalization error and, empirically, can compensate for distribution shifts, sometimes even outperforming the unadjusted (train-Bayes-matched) baseline.

**Strengths:**

1. The paper is clearly written and well structured, with a precise statement of the problem and motivation.

2. The derivation of a closed-form expression for the ICL generalization error appears technically sound and provides theoretical insight.

3. The study of how pretraining affects in-context learning behavior is an important and timely question that the paper addresses thoughtfully.

**Weaknesses:**

1. **Limited practical applicability of the optimal temperature:** The theoretically derived optimal temperature $ \tau_{\text{opt}} $ depends on quantities from the true test distribution (e.g., $ \Sigma_x^{\text{test}}, \Sigma_w^{\text{test}} $), which are unknown in practice.  Therefore, the practical takeaway essentially reduces to tuning $ \tau $ by sweeping or validation, which is intuitive and already common.  The paper could better clarify whether any *predictive or data-dependent heuristic* for $ \tau $ can be inferred from their theory.

2. **Clarifying the intuition about shift magnitude and temperature:**  I would like to ask the authors whether the following intuition is consistent with their results:
 – *If the distribution shift is small, the optimal $ \tau $ should remain close to 1. if the shift is large (e.g., much higher input variance or noise), $ \tau $ should increase accordingly.*  This seems qualitatively aligned with current LLM practices, where higher attention temperature can stabilize performance under distribution mismatch. This seesm to be the case in Figure 2, but how can one see this from equation 12? Any insights?

3. **Constructed vs. learned parameters:** The entire analysis relies on constructed parameters (Lemma 4.1) rather than parameters obtained by training, even in the simplified model. It remains unclear whether training via gradient descent would converge to these analytic forms or perhaps to different parameters that perform better than the train-Bayes estimator due to implicit biases of optimization. Since attention models are highly expressive, this reliance on a fixed construction limits how directly the conclusions translate to actual trained models.

4. **Relation to test-time adaptation (TTT):**  Allowing temperature adjustment at test time is essentially a *restricted form* of test-time tuning or adaptation (TTT).  Naturally, permitting any adaptation using in-context examples can improve performance; in this work, the authors allow only a single scalar ($ \tau $) to change.  In principle, broader adaptation (e.g., re-scaling attention weights) could approximate the *test-Bayes optimal* predictor.  This connection to TTT could be discussed more explicitly.

**Questions:**

See weaknesses.

---

> ### Author Response · Authors · 2025-11-19
>
> We thank the reviewer for the thoughtful and constructive feedback. We appreciate your recognition that the paper is clearly written, that the closed-form characterization of the ICL generalization error is technically sound and insightful, and that the work addresses an important and timely question. Before addressing the detailed points, we would like to emphasize that the central contribution of the paper is theoretical: we provide **the first closed-form characterization of ICL generalization and optimal attention temperature under distribution shift for a softmax-like attention mechanism, from which all practical insights—including our temperature-selection heuristic—naturally follow**. We hope this framing helps clarify the intended scope of the work as we address your comments below.
>
> > Limited practical applicability of the optimal temperature: $\dots$ The paper could better clarify whether any predictive or data-dependent heuristic for $\tau$ can be inferred from their theory.
>
> Thank you for the comment. In fact, **our theory *does* yield a predictive, data-dependent heuristic for choosing $\tau$. As detailed in Section 5.2 and Appendix J, Theorem 4.7 implies that the optimal temperature is roughly proportional to the *variance-to-mean ratio of the pre-softmax attention scores*.** This statistic can be estimated directly from data, without access to the true test distribution. We apply this heuristic in our LLM experiments (Figure 3), where it consistently improves robustness under distribution shift.
>
>
> > Clarifying the intuition about shift magnitude and temperature: I would like to ask the authors whether the following intuition is consistent with their results:
> – If the distribution shift is small, the optimal $\tau$ should remain close to 1. if the shift is large (e.g., much higher input variance or noise), $\tau$ should increase accordingly. This seems qualitatively aligned with current LLM practices, where higher attention temperature can stabilize performance under distribution mismatch. This seesm to be the case in Figure 2, but how can one see this from equation 12? Any insights?
>
> Thank you for requesting clarification. *The intuition you describe is indeed fully consistent with both our theoretical and empirical results*. As discussed below Theorem 4.7, consider the example where the input distribution shifts from $\mathcal{N}(0, I)$ to $\mathcal{N}(0, cI)$. In this case, the optimal temperature (Eq. 15) simplifies to $\tau_{\mathrm{opt}} = c$ for large $l$, showing explicitly that larger input-variance shifts require proportionally larger $\tau$. Likewise, increases in noise level also raise the optimal temperature, since the noise variance appears in the numerator of $F_1$ in Eq. 15. Thus, Equation (12-15) encodes the same principle: when the variance of pre-softmax attention score grows relative to its mean, the optimal $\tau$ increases accordingly. **To facilitate the interpretation of the closed-form equation (Eq. 15) for optimal attention temperature, we added a new section (Appendix M) at the end of the paper during the rebuttal. It provides a reduction (simplification) of Eq. 15 (optimal temperature) for a simplified case and demonstrates how the closed-form optimal temperature changes with respect to various distribution shifts *by providing new analytical and experimental results*.**

---

> > ### Author Response · Authors · 2025-11-19
> >
> > > Constructed vs. learned parameters: The entire analysis relies on constructed parameters (Lemma 4.1) rather than parameters obtained by training, even in the simplified model. It remains unclear whether training via gradient descent would converge to these analytic forms or perhaps to different parameters that perform better than the train-Bayes estimator due to implicit biases of optimization. Since attention models are highly expressive, this reliance on a fixed construction limits how directly the conclusions translate to actual trained models.
> >
> > We appreciate the opportunity to clarify this point. **As highlighted in Remark 4.3, our characterization of ICL performance (Theorem 4.6) and the optimal temperature (Theorem 4.7) *does not* rely on the specific constructed parameters of Lemma 4.1**. These results hold under the more general assumption stated in Assumption 4.5, which is independent of any particular analytic construction and is compatible with parameters obtained by training.
> >
> > Lemma 4.1 serves a different purpose: it provides an explicit parameter setting (mimicking the Bayes-optimal model) that allows us to cleanly study the effect of controlled distribution shifts in Section 4.2. The empirical results with GPT-2 and LLaMA further indicate that our theoretical predictions (selection and benefits of optimal attention temperature) naturally arise in real-world trained models.
> >
> >
> > > Relation to test-time adaptation (TTT): Allowing temperature adjustment at test time is essentially a restricted form of test-time tuning or adaptation (TTT). Naturally, permitting any adaptation using in-context examples can improve performance; in this work, the authors allow only a single scalar ($\tau$) to change. In principle, broader adaptation (e.g., re-scaling attention weights) could approximate the test-Bayes optimal predictor. This connection to TTT could be discussed more explicitly.
> >
> > Thank you for pointing out this connection. Although temperature adjustment can be viewed as a very restricted form of test-time adaptation (TTT), our approach differs from standard TTT: **we *do not* perform any optimization or fine-tuning at test time**. Instead, we select a single scalar $\tau$ using the closed-form equation or insights from our analysis (Theorem 4.7, Appendix J). This yields a simple and interpretable form of adaptation without the complexity of typical TTT procedures.
> >
> > ---
> >
> > Overall, our work provides the first theoretical foundation for understanding optimal attention temperature in in-context learning under distribution shifts, including exact analytical expressions that characterize both ICL performance and the optimal temperature. **During the rebuttal phase, we introduced new analytical and experimental results in Appendix M that enhanced the interpretability of our theoretical results (Theorems 4.6–4.7). In parallel, our rebuttal comments provided additional clarity on the theoretical contribution and novelty of the paper.** We believe we have fully addressed the concerns you raised, and we respectfully ask you to reconsider your evaluation in light of these clarifications.

---

> > > ### Author Response · Authors · 2025-11-26
> > >
> > > Dear Reviewer mUBM, we would like to kindly follow up on your review and ask whether our clarifications regarding the practical variance-to-mean heuristic for choosing the temperature, the constructed-vs-learned parameters, and the relation to test-time adaptation (including the new material in Appendix M) resolve your concerns. If there is anything still missing or unclear, we would greatly appreciate additional questions or an updated assessment.

---

### Official Review · Reviewer_fpkd · 2025-11-01

**Soundness:** 3
**Presentation:** 4
**Contribution:** 2
**Rating:** 4
**Confidence:** 3

**Summary:**

This paper studies in context learning using a stylized model (common in the literature) showcasing the ability of an attention layer to learn linear functions "in-context". The paper deviates from prior work by considering a ridge-regression type Bayes-optimal predictor (incorporating a gaussian prior on the unknown parameter) and a "linearized" softmax attention. The paper compars the performance of the optimal transformer, and considers the degradation of this performance under distribution shift (a known failure mode of ICL amongst softmax/linear attention), finally demonstrating that tuning the temperature parameter in the attention mechanism can help alleviate some of the said degradation.

**Strengths:**

The Llama/GPT-2 experiment is particularly interesting. It seems to have prescriptive advice for extracting the optimal behaviour from softmax-attention - by setting the temperature to be related to the ratio of the variance of the pre-softmax scores to the mean of those scores.

**Weaknesses:**

Showing that linearized-softmax can express something similar to the Bayes optimal predictor seems a little derivative of a similar analysis for linear attention.

Theorem 4.7 is difficult to parse, I think it would help if it could be written out in terms of the optimal parameter values from Lemma 4.1. I think something like this happens in Appendix J.

**Questions:**

I didnt understand the set-up of the "real world" experiments. Are the optimal temperatures calculated per attention layer per batch? What is special about ICL that this rule should only be applied in this setting (this doesnt seem to be related to any distribution shift)? I am giving a slightly lower score now. Once I have some more clarity on section 5.2 and how this can be applied to real models I will likely raise it. I apologize if my question has been addressed already in the paper.

Is there an interpretation of post-softmax attention weights with the optimal temperature? An example of the type of statement i would hope for is that that they are approximately a standard log-normal under some assumption about the pre-softmax distribution or that there is some gap between the highest and average post-softmax attention weights, etc.

Comparing Figure 6 to Figure 3(a). How is the non-monotonicity of Exact Match Score for optimal temperature to be compared to the monotonicity of gpt-2 for regression problems with optimal temperature?

---

> ### Author Response · Authors · 2025-11-19
>
> We thank the reviewer for the thoughtful comments and for the interest in our theoretical and empirical findings. We are glad that you found the presentation excellent and appreciated the practical value of the LLaMA/GPT-2 experiments. Before addressing the specific questions, we would like to clarify that the paper’s contributions are primarily theoretical: we derive the first analytical expressions for ICL generalization error and for the optimal attention temperature under distribution shift, with experiments included solely to illustrate these theoretical insights. We hope this framing helps contextualize our responses below.
>
> > Showing that linearized-softmax can express something similar to the Bayes optimal predictor seems a little derivative of a similar analysis for linear attention.
>
> Our contribution goes beyond prior linear-attention analyses (e.g., Zhang et al. (2024)). Specifically:
> * **As opposed to linear attention, the linearized-softmax attention we consider can capture temperature effects similar to softmax (please see Figure 5 in Appendix D for concrete evidence)** while remaining analyzable. This motivates our focus on linearized-softmax attention, which, to the best of our knowledge, has not been studied in the literature.
> * The additional terms in **the linearized-softmax add extra complexity to our precise analysis of (ICL) generalization error** in comparison to the prior literature on linear attention.
> * We **relax many assumptions (e.g., on data distribution) utilized in the prior analyses** to make the setting more general, such that it captures a wide range of distribution shifts and their relation to optimal temperature. This also extends the theoretical challenges of our precise analysis of (ICL) generalization error.
> * Our theoretical results (Theorems 4.6–4.7) provide closed-form equations for the generalization error and **optimal attention temperature that minimizes generalization error** under distribution shift—a setting completely **absent in prior linear-attention analyses**.
>
> We will clarify this distinction at the start of Sec. 3.3 by adding one sentence: "Unlike previous analyses of purely linear attention, our linearized-softmax retains temperature-dependent effects and thus enables deriving an optimal temperature, which has no analogue in prior linear-attention models."
>
> > Theorem 4.7 is difficult to parse, I think it would help if it could be written out in terms of the optimal parameter values from Lemma 4.1. I think something like this happens in Appendix J.
>
> We appreciate the reviewer's suggestion. As noted above, our theoretical setting is intentionally general to capture the interplay among temperature, in-context learning, and distribution shift. Consequently, the resulting analytical forms for the ICL error (Theorem 4.6) and the optimal temperature (Theorem 4.7) are necessarily intricate. Further simplification is generally infeasible without imposing restrictive assumptions. We have already discussed an example case immediately below Theorem 4.7, illustrating the behavior of optimal temperature. However, we agree with you in the sense that the discussion of the optimal temperature can be extended.
>
> **To further enhance the interpretability of the optimal temperature (Theorem 4.7), we added a new section (Appendix M) *supplying new analytical and experimental results* to the end of the paper during the rebuttal period**. In this new section, we both analytically and experimentally demonstrate how the optimal attention temperature (as stated in Theorem 4.7) varies for a simple yet comprehensive-enough distribution shift scenario that encompasses shifts in inputs, tasks, or noise. Furthermore, the discussion in this section is also connected to the moment-ratio heuristic (Appendix J), which we used in our LLM experiments, thereby strengthening the association between our setting and the moment-ratio heuristic that can be utilized in practice. We believe these explanations and examples make the theoretical results as transparent as possible within the scope of our framework.

---

> > ### Author Response · Authors · 2025-11-19
> >
> > > I didnt understand the set-up of the "real world" experiments. Are the optimal temperatures calculated per attention layer per batch?
> >
> > We appreciate the opportunity to clarify this point. In our LLM experiments, we fix a single scalar $\tau$ and set the attention temperature of each layer to $\tau \times \sqrt{d_k}$, where $d_k$ denotes the key dimension of that layer, ensuring dimension independence. For the *optimal temperature* setting, $\tau$ is estimated once using the *variance-to-mean ratio of the pre-softmax scores* (as motivated by Theorem 4.7 as explained in Appendix J), computed over the entire validation dataset.
> >
> > While computing layer- or batch-specific optimal temperatures is an interesting direction, it is nontrivial due to the high variance and instability of such per-layer estimates. We therefore leave this for future work. Importantly, starting from our single-layer theoretical analysis, we show that a single global $\tau$ already yields consistent improvements in ICL performance and robustness under distribution shift.
> >
> > > What is special about ICL that this rule should only be applied in this setting (this doesnt seem to be related to any distribution shift)?
> >
> > Thank you for the insightful question. To the best of our knowledge, **the role of attention temperature in *in-context learning* has not been theoretically or empirically studied** in prior work. Our focus on ICL is therefore motivated by a clear gap in the literature: ICL relies on a frozen model whose adaptation occurs *entirely through attention*, making temperature a direct and influential mechanism. This is precisely the setting in which we can cleanly characterize how temperature interacts with distribution shift and derive an optimal value.
> >
> > At present, our theoretical results are specific to this ICL framework and its associated assumptions. Extending the analysis to other training or inference paradigms (e.g., fine-tuning, instruction tuning, or supervised tasks) would require substantially different modeling choices and is an interesting direction for future work. Our current study demonstrates that even within the ICL setting, optimal temperature tuning has a clear and measurable robustness benefit.
> >
> >
> > > I am giving a slightly lower score now. Once I have some more clarity on section 5.2 and how this can be applied to real models I will likely raise it. I apologize if my question has been addressed already in the paper.
> >
> > Thank you for the kind comment and for indicating that clarifications on Section 5.2 may raise your score. The practical guideline derived from our theory is that the optimal temperature is approximately proportional to the ratio between the second moment and the first moment of the pre-softmax attention scores (as detailed in Appendix J). This rule is directly applied in our LLM experiments in Section 5.2: we estimate these moments once on a validation set and use the resulting scalar temperature across all attention layers.
> >
> > We hope this clarification demonstrates how our theoretical insight translates into a concrete and easily implementable procedure for real models. We would be happy to provide any additional details if helpful.

---

> > > ### Author Response · Authors · 2025-11-19
> > >
> > > > Is there an interpretation of post-softmax attention weights with the optimal temperature? An example of the type of statement i would hope for is that that they are approximately a standard log-normal under some assumption about the pre-softmax distribution or that there is some gap between the highest and average post-softmax attention weights, etc.
> > >
> > > Thank you for this insightful question. In our setting, the pre-softmax attention scores are approximately Gaussian, implying that the post-softmax weights follow a log–normal–type distribution, whose spread is controlled by the temperature. As illustrated in Appendix D (Figure 5), *varying the temperature primarily modulates the variance of the post-softmax weights*, while the overall shape remains closely tied to the distribution of the pre-softmax scores.
> > >
> > > The optimal temperature derived in Theorem 4.7 balances this variance against the mean score gap, preventing attention collapse and yielding a stable ratio between the largest and average attention weights. We will highlight this interpretation more explicitly in the final version.
> > >
> > > > Comparing Figure 6 to Figure 3(a). How is the non-monotonicity of Exact Match Score for optimal temperature to be compared to the monotonicity of gpt-2 for regression problems with optimal temperature?
> > >
> > > Thank you for raising this point. The apparent *discrepancy arises from the fundamentally different nature of the two settings*. In the GPT-2 regression experiment (Figure 3(a)), we evaluate a controlled linear regression task with a simple and well-specified distribution shift (changing the input covariance from $I$ to $3I$). In this regime, the optimal temperature directly counteracts the shift and yields a monotonic improvement in ICL error with context length.
> > >
> > > In contrast, the LLM experiment in Figure 6 involves real-world text data with label-level distribution shift and substantially more complex attention dynamics. In this setting, we do not expect a single global temperature parameter (shared across all layers) to fully eliminate non-monotonicity in Exact Match accuracy. Nonetheless, as Figure 6 shows, applying the theoretically motivated temperature still produces consistent robustness gains, even if perfect monotonicity is not attainable in such a complex setting.
> > >
> > > ---
> > >
> > > Overall, **our paper presents the first foundational theoretical results demonstrating both the existence of an optimal attention temperature for ICL and its practical benefits in improving robustness under distribution shift while providing insights regarding the choice of optimal attention temperature**. During the rebuttal period, the new analytical and experimental results added in Appendix M further improved the interpretability of our theoretical findings (Theorems 4.6–4.7), and our rebuttal comments clarified the theoretical contribution and novelty of the work. We hope that the clarifications above clearly address your concerns, and we would be grateful if you would reconsider your evaluation in light of these points.

---

> > > > ### Author Response · Authors · 2025-11-26
> > > >
> > > > Dear Reviewer fpkd, we would like to briefly follow up to ask whether our clarifications on Sec. 5.2 and the additional analytical/experimental results in Appendix M address your questions about the LLM experiments and the interpretation of the optimal temperature. If any part remains unclear, we would be very grateful for further feedback and for your reconsideration of the current evaluation.

---

### Author Response · Authors · 2025-11-30

Dear Area Chair,

Given the recent system changes and the early termination of the discussion phase, we would like to provide a concise summary of our paper’s contributions and how our rebuttal and additional material address the reviewers’ concerns.

Our work offers, to our knowledge, the first **closed-form characterization of ICL generalization error and the corresponding optimal attention temperature under distribution shift for softmax-like attention** (Theorems 4.6–4.7). Unlike prior analyses focused on linear attention, our approach (i) works with a linearized softmax that preserves genuine temperature-dependent behavior (as illustrated in Appendix D), and (ii) explicitly separates pretraining and test-time distribution parameters. This allows us to rigorously analyze how shifts in input covariance, task distribution, and noise affect ICL, and how appropriately chosen temperatures can mitigate these effects.

During the rebuttal, we (1) **clarified that the primary contribution is theoretical**—not an empirical tuning recipe; (2) explained that attention temperature in this ICL context has not been previously analyzed; (3) showed that the main theorems rely only on Assumption 4.5, not on any specific parameter constructions from Lemma 4.1; and (4) added a **new Appendix M containing additional analytical and experimental results** illustrating the behavior of Eq. (15) and the dependence of the optimal temperature on different types and magnitudes of distribution shift. We also clarified how these results motivate a practical **variance-to-mean heuristic** for temperature selection, which is supported by our LLaMA2-7B experiments.

Reviewers generally agreed that the paper is technically sound, clearly presented, and tackles a timely problem. One reviewer also indicated that our clarifications on the LLM experiments and practical temperature choice would **likely lead to a higher score, had the discussion period continued**. We believe that our responses and the new Appendix M substantially address the remaining concerns regarding novelty, practical guidance, and the link between the simplified model and real LLMs. We respectfully hope that you will take this into account when forming your recommendation.

We are aware of the additional burden placed on ACs by the recent changes, and we hope this short summary is helpful in your assessment.

Best regards,

Authors

---

### Meta-Review · Area_Chair_5RTL · 2026-01-06

**Summary:**

This work analyzes the optimal attention temperature in a transformer with a linearized softmax model during ICL. Closed form expression of the optimal temperature is derived, and verified through experiments.

**Reviewer Concerns:**

- The analysis of linearized softmax is a minor extension of prior work on linear attention.
- The theory relies on analytically constructed parameters instead of parameters obtained through training. It is unclear whether gradient-based optimization would converge to these solutions.
- The theoretically optimal temperature depends on unknown test-distribution quantities, which limits the applicability of the results in practice.
- The results are hard to interpret, as it does not clearly characterize how generalization error or optimal temperature change under distribution shifts,

The authors addressed the last two concerns during the rebuttal. However, the novelty and technical contribution of this work seems limited.

**Reviewer Scores:**

The original scores were 4/4/4/4. None of the reviewers responded during the rebuttal. They would likely remain their scores.

---

### Decision · Program_Chairs · 2026-01-26

Reject